# Self-generated surface magnetic fields inhibit laser-driven sheath acceleration of high-energy protons

M. Nakatsutsumi [1,2,4], Y. Sentoku[3,5], A. Korzhimanov [6], S.N. Chen[1,6], S. Buffechoux[1], A. Kon[3,7,10], B. Atherton[8], P. Audebert[1], M. Geissel[8], L. Hurd [1,11], M. Kimmel[8], P. Rambo[8], M. Schollmeier [8], J. Schwarz[8], M. Starodubtsev[6], L. Gremillet[9], R. Kodama[3,4,7] & J. Fuchs [1,6]

High-intensity lasers interacting with solid foils produce copious numbers of relativistic electrons, which in turn create strong sheath electric fields around the target. The proton beams accelerated in such fields have remarkable properties, enabling ultrafast radiography of plasma phenomena or isochoric heating of dense materials. In view of longer-term multidisciplinary purposes (e.g., spallation neutron sources or cancer therapy), the current challenge is to achieve proton energies well in excess of 100 MeV, which is commonly thought to be possible by raising the on-target laser intensity. Here we present experimental and numerical results demonstrating that magnetostatic fields self-generated on the target surface may pose a fundamental limit to sheath-driven ion acceleration for high enough laser intensities. Those fields can be strong enough ($\sim 10^5$ T at laser intensities $\sim 10^{21}$ W cm$^{-2}$) to magnetize the sheath electrons and deflect protons off the accelerating region, hence degrading the maximum energy the latter can acquire.

[1] LULI—CNRS, École Polytechnique, CEA: Université Paris-Saclay; UPMC Univ Paris 06: Sorbonne Universités, Palaiseau cedex F-91128, France. [2] European XFEL, GmbH, Holzkoppel 4, 22869 Schenefeld, Germany. [3] Institute of Laser Engineering, Osaka University, Suita, Osaka 565-0871, Japan. [4] Open and Transdisciplinary Research Initiatives, Osaka University, Suita, Osaka 565-0871, Japan. [5] Department of Physics, University of Nevada, Reno, Nevada 89557, USA. [6] Institute of Applied Physics, 46 Ulyanov Street, 603950 Nizhny Novgorod, Russia. [7] Graduate School of Engineering, Osaka University, Suita, Osaka 565-0871, Japan. [8] Sandia National Laboratories, Albuquerque, NM 87123, USA. [9] CEA, DAM, DIF, Arpajon F-91297, France. [10] Present address: Japan Synchrotron Radiation Research Institute, Sayo, Hyogo 679-5198, Japan. [11] Present address: Department of Physics and Astronomy, Clemson University, Clemson, SC 29634, USA. Correspondence and requests for materials should be addressed to M.N. (email: motoaki.nakatsutsumi@xfel.eu) or to J.F. (email: julien.fuchs@polytechnique.fr)

In the search to increase the energy of proton and ion beams accelerated by intense lasers from solid targets[1], significant work has been devoted to investigating the ion energy scaling with laser[2–9] and target[10–18] parameters (see also Supplementary Note 4). In the frame of the most investigated acceleration mechanism, i.e., target-normal-sheath-acceleration (TNSA), which is driven by laser-generated electrons, empirical formulas and analytical models[3–5,7,19–22] have been used to obtain the sought-after scaling. These models have been found adequate for $I_L\lambda_L^2 < 10^{20}$ W $\mu m^2 cm^{-2}$ (where $I_L$ and $\lambda_L$ are the laser intensity and wavelength, respectively)[2,4,23], but they appear to severely overestimate the measured proton energies at higher intensities[5]. This discrepancy has been attributed to their reduced geometry (usually 1D) or to simplistic assumptions about the plasma dynamics (e.g., isothermal or adiabatic). By contrast, particle-in-cell (PIC) numerical codes provide a first-principles description of the laser-driven ion acceleration process[24,25], yet computational constraints restrict current simulations to rather limited spatiotemporal domains and/or reduced dimensionality. Despite these shortcomings, there has been anticipation of exceeding the 100 MeV energy threshold via TNSA at laser intensities in the $10^{21}$ W $\mu m^2 cm^{-2}$ range.

Most of these studies, however, have overlooked a potentially important factor: the feedback effect on the electrons and accelerating ions of magnetic (B) fields that are self-generated on the target surfaces[25] and can act detrimentally on the particle dynamics for high enough laser intensities. Recently, mounting experimental evidence[26–30] has been obtained showing that tens of MegaGauss (MG) strength B-fields grow in a few 100 fs[28,29] on the target surfaces for $I_L\lambda_L^2 \geq 10^{19}$ W $\mu m^2 cm^{-2}$. Moreover, we have shown that, for laser pulses with a high temporal contrast, a condition that is sought in order to irradiate ultrathin targets, and hence increase the electron sheath density and the accelerated ion energy[6,9,10,11,15], the surface B-fields are maintained over durations (tens of ps) much longer[29] than the timescale of energy transfer from the electrons to the ions (typically < 1 ps[4,19]). The potentially detrimental effect of the B-fields on ion acceleration had been evoked in a 3D PIC simulation study[25], but up to now little attention has been paid to it, likely because the MG-strength B-fields observed at present-day laser intensities do not indeed impact ion acceleration.

Here we show that when irradiating targets beyond $10^{20}$ W $\mu m^2 cm^{-2}$ in laser intensity, which is here achieved using a tightly focused laser[31], B-fields of the order of 100 MG ($10^4$ T) can grow on the target surfaces. Inductive in nature, they arise from the steep transverse gradient of the sheath electric field accelerating the ions, and hence may continuously affect the cloud of electrons and ions expanding from the target. At even higher laser intensities ($10^{21}$ W $\mu m^2 cm^{-2}$), these B-fields can grow to the Giga-Gauss (GG) level. The electrons subjected to such extreme fields become trapped on the target surface, where they further undergo an $\mathbf{E} \times \mathbf{B}$ drift away from the sheath axis. The resulting inhibition of the electron forward motion is found to hamper the proton acceleration. In addition, part of the protons is significantly deflected outwards, thus degrading the proton beam's exceptional emittance[1]. But, we also find that using very short laser pulses (a few tens of femtoseconds) might be a way to mitigate the magnetic inhibition effect under consideration since, in such conditions, the protons can be accelerated promptly enough before the electron trajectories are strongly perturbed by the B-fields. The magnetization effect highlighted in our study should be carefully considered when designing ion sources at the upcoming multi-PW laser facilities[32], in particular those with >100 fs duration[33]. Indeed, extrapolating the existing empirical or theoretical scalings to high-intensity conditions may lead to considerable overestimation of the proton energies, especially for not ultrashort laser pulses.

## Results

**Magnetic fields in relativistic laser–solid interaction.** Figure 1 shows the spatial distribution of the quasistatic B-fields ($B_z$) observed in 2D PIC simulations (see Methods) performed at peak laser intensities $I_L\lambda_L^2 = 6.5 \times 10^{19}$ W $\mu m^2 cm^{-2}$ (Fig. 1a) and $2 \times 10^{21}$ W $\mu m^2 cm^{-2}$ (Fig. 1e). These fields mainly develop on the target surfaces and are polarized normal to the 2D simulation plane. In an actual 3D geometry, as already observed[28,29], they are toroidal and oriented clock-wise around the target normal. Their transverse profile presents a steep increase in strength towards the center, up to the edge of the narrow central region where it abruptly vanishes and changes sign. They are stronger at the rear surface compared to the front, consistently with our previous measurements also performed at high temporal contrast[29], and they reach strengths of about 100 MG and 500 MG, respectively (see also Fig. 2a). There, the weak-field region occupies only a small fraction of the ~40 $\mu m$ transverse extent of the electron sheath[34].

These magnetostatic fields are predominantly driven by the currents associated with the laser-driven hot electrons[35–37], the same ones responsible for building up the ion accelerating sheath (the protons being preferentially accelerated due to their lowest charge-to-mass ratio). The hot electrons are injected into the target with kinetic energies of several MeV for the laser intensities considered here (see Methods). When exiting the target, a small fraction of them escape into the vacuum[38], but most are retained by the electrostatic potential barrier, and hence form sheaths on the target surfaces[1]. The inductive generation of the B-field is mainly determined by the spatio-temporal variations of the longitudinal sheath field: $\partial B_z/\partial t \sim \partial E_x/\partial y$[39] (see Methods). For the sake of simplicity, and consistently with the 2D PIC simulations, this equation is written in 2D geometry. The B-field rapidly grows during the laser irradiation and eventually saturates, an upper limit being reached when the magnetic and electron pressures become comparable: $B_{max}^2/2\mu_0 \approx n_{h,rear}k_B T_0$, where $T_0$, $n_{h,rear}$ are the hot electrons' initial temperature and density (at the target rear), $k_B$ is the Boltzmann constant and $\mu_0$ denotes the vacuum permeability. This should be considered as an upper limit since, as soon as the electrons become magnetized, a further rise in the B-field strength requires a similar rise in $n_{h,rear}$, which is increasingly difficult to achieve. This scaling predicts that $B_{max}$ can exceed 0.5 GG at laser intensities $\geq 10^{21}$ W $\mu m^2 cm^{-2}$, which is supported by the simulation shown in Fig. 1e (see also Fig. 2a).

**Model for plasma expansion and magnetic-field generation.** To determine in which measure the B-fields depicted in Fig. 1a, e are detrimental to proton acceleration from the target rear, we resort to a simple 1D analytical model of the temporal evolution of the particles and fields (see Methods). The use of a 1D model is justified provided that the acceleration length of the protons does not exceed the lateral extent of the sheath[23]. The electron temperature, $T_e(t)$, is assumed to evolve as $T_e(t) = T_0$ during $0 \leq t \leq \tau_L$ and $T_e(t) = T_0(\tau_L/t)^2$ for $t > \tau_L$ to mimic adiabatic cooling[20]. The model yields the longitudinal electrostatic field, $E_x(t)$, from which we evaluate the magnetostatic field, $B_z(t)$, as well as the proton front position, $x_{front}(t)$, and the maximum ion velocity, $v_p(t) = \dot{x}_{front}(t)$. These quantities are plotted in Fig. 1b, f. The magnetization level can be assessed from comparison of $x_{front}(t)$ with the typical values of the electron and proton Larmor radii, $R_L^{e,p}(t)$ (see Methods): the particles can be considered strongly magnetized in the cloud if $R_L^{e,p}/x_{front} < 1$. For a laser intensity of $6.5 \times 10^{19}$ W $\mu m^2 cm^{-2}$, Fig. 1c shows that the electrons become magnetized in ~100 fs after the start of the plasma expansion (and during the laser pulse irradiation). When increasing the laser

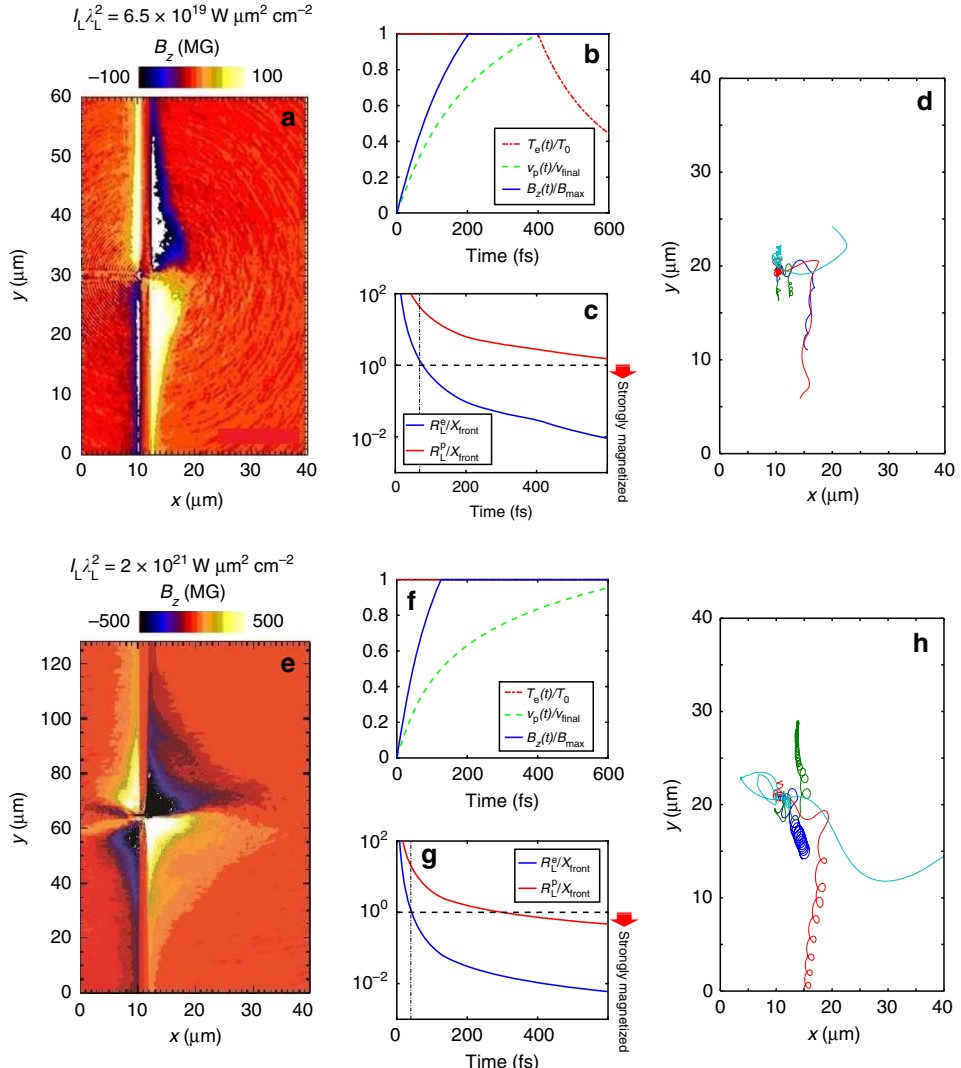

**Fig. 1** Laser-driven magnetic-field generation and resulting particle dynamics. **a, d, e, h** 2D particle-in-cell (PIC) simulation results obtained using the (**a, e**) PICLS[26] and (**d, h**) PICADOR[59] codes (see Methods). The laser pulse impinges from the left onto a 2-μm thick Al foil, coated on its rear side (located at $x$ = 12 μm) with a 20-nm thick proton layer. The laser FWHM spot size $\phi_L$, duration $\tau_L$, wavelength $\lambda_L$ and intensity $I_L\lambda_L^2$ are, respectively, (**a, d**) 1 μm, 400 fs, 0.5 μm, $6.5 \times 10^{19}$ W μm$^2$ cm$^{-2}$ and (**e, h**) 1.6 μm, 700 fs, 1 μm and $2 \times 10^{21}$ W μm$^2$ cm$^{-2}$. **a, e** Magnetostatic field $B_z$ (in MG) developing inside and outside the target at 100 fs after the laser peak. **d, h** Sample electron trajectories from the PIC simulations, exiting the target at the laser peak. In **d**, the electron energies lie in the same range: 13.8 MeV (green), 14.3 MeV (red), 12.3 MeV (blue) and 18.7 MeV (cyan), yet the green electron proves more strongly magnetized because it is ejected into the vacuum about 50 fs later, and thus experiences a higher $B$-field. In **h**, the electron energies are 25.2 MeV (blue), 27.7 MeV (green), 90.9 MeV (red) and 162.7 MeV (cyan). **b, c, f, g** Results from the 1D expansion model (see Methods) at a laser intensity of (**b, c**) $I_L\lambda_L^2 = 6.5 \times 10^{19}$ W μm$^2$ cm$^{-2}$ and (**f, g**) $I_L\lambda_L^2 = 2 \times 10^{21}$ W μm$^2$ cm$^{-2}$ using the corresponding PIC simulation parameters. **b, f** Time evolutions of the proton velocity, $\nu_p(t)$, normalized to its final value (dashed green line), of the electron temperature, $T_e(t)$, normalized to its initial value ($T_0 = 1.1$ MeV in **b** and $T_0 = 5.6$ MeV in **f**, see Methods) (dashed-dotted red line), and of the inductive $B$-field, normalized to its predicted saturation value $B_{max} \equiv (2\mu_0 n_{h,rear} k_B T_0)^{1/2}$ ($B_{max} = 144$ MG in **b** and $B_{max} = 537$ MG in **f**) (solid blue line). **c, g** Time evolutions of the electron ($R_L^e$, blue) and proton ($R_L^p$, red) radii, normalized to the instantaneous longitudinal extent of the proton plasma, $x_{front}(t)$. The horizontal dashed black line delimitates the boundary between the regimes of strong ($R_L^{e,p}/x_{front} < 1$) and weak ($R_L^{e,p}/x_{front} > 1$) magnetization. The vertical dashed-dotted line indicates the time when the ion front has moved a distance larger than the local Debye length

intensity to $2 \times 10^{21}$ W μm$^2$ cm$^{-2}$ as shown in Fig. 1g, not only do the electrons get strongly magnetized even more quickly, but the protons turn out to be strongly magnetized too during the laser pulse. While this 1D analytical model does not integrate the feedback of the magnetization on the particle cloud expansion and proton acceleration, it clearly indicates that $B$-fields are likely to impact the electron and proton dynamics at high laser intensity. Complementarily, Fig. 1d, h shows the trajectories of sample electrons in the PIC simulation for the two intensities considered (see also Supplementary Note 2, Supplementary Fig. 2,

Supplementary Fig. 3 for modeling of electron trajectories). In agreement with the model predictions, these trajectories show that the electrons are magnetized, drifting away from the center of the sheath, due to the combined actions of the $B$-field gradient and the $\mathbf{E} \times \mathbf{B}$ drive, the effect being aggravated with the laser intensity.

The main effect of the $B$-field on the electrons is, by scattering them outward along the target surface, to reduce the longitudinal electron pressure and density (see also Supplementary Note 1 and Supplementary Fig. 1), which shortens the electron sheath and

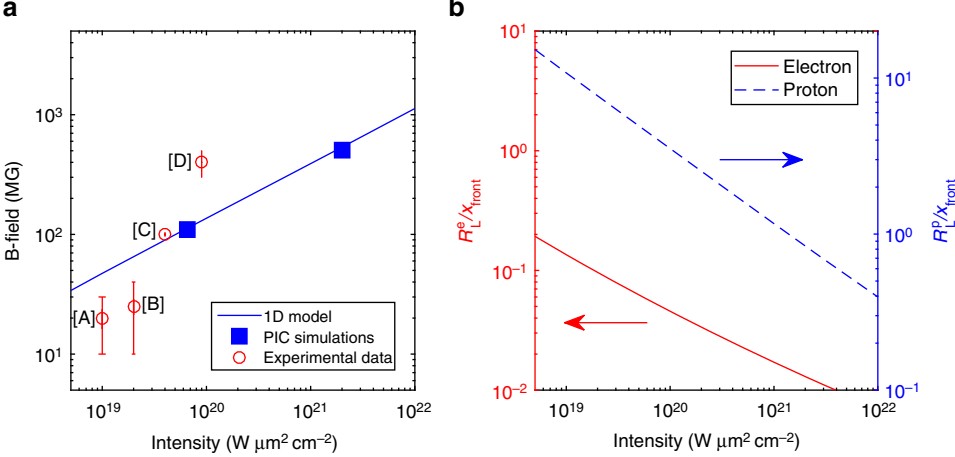

**Fig. 2** Magnetic-field strength and particle magnetization vs. laser intensity. **a** Blue solid curve: magnetic field strength (in MG) as a function of laser intensity ($I_L \lambda_L^2$) from the 1D model (see text for details on the parameters) taken at 350 fs after the plasma expansion starts (i.e., corresponding to the temporal peak for the 700 fs pulse). Experimental data are shown as red points ([A] correspond to ref. [27], [B] to ref. [29], [C] to ref. [28], and [D] to ref. [30]), while PIC simulation results are shown as blue boxes (see Fig. 1). **b** Normalized Larmor radii of electrons ($R_L^e / x_{front}$, red) and protons ($R_L^p / x_{front}$, dashed blue), at the laser intensity peak, as predicted by the 1D model. The laser parameters are those of Fig. 1e–h: $\phi_L = 1.6\,\mu m$, $\tau_L = 700\,fs$ and $\lambda_L = 1\,\mu m$

decreases the $E$-field strength. As for the protons, they tend to be deflected outward, i.e., toward lower-sheath-field regions, also hampering the energy gain they could expect. Overall, the ion acceleration process is changed from a quasi-1D geometry (neglecting the intrinsic divergence of the hot electrons, which is valid during most of the ion acceleration phase[23]) into a pronounced, less efficient, 3D one. Another expected detrimental effect of the $B$-field is the inward force exerted by the fluid-like magnetic pressure on the target surface, which tends to counter-act the accelerating TNSA field.

**Laser intensity dependence of magnetization effects**. Figure 2 quantifies the impact of the magnetic field on the particle dynamics as a function of the laser intensity, for fixed values of the laser spot size (1.6 µm), duration (700 fs) and wavelength (1 µm). Figure 2a plots the $B$-field strength vs. laser intensity as predicted by our 1D model. This theoretical scaling is consistent with the PIC simulation results presented above. Also plotted are experimental data taken from refs [27–30]. Note that these data are not all acquired using the same spot size, but as detailed in Supplementary Note 6 and Supplementary Fig. 7, the dependence of the $B$-field strength on the laser focal spot at a given intensity is actually rather weak. This is due to the fact that the sheath transverse gradient (responsible for the $B$-field generation) is dominated by fast recirculation and transverse spread of the hot electrons before the ions have time to move. This holds as long as the laser spot size is smaller than the transverse sheath size (of the order of 40 µm, Fig. 1a, e and Supplementary Note 6). Figure 2b plots the particle magnetization parameter at the time of the laser peak, as calculated using the same 1D model as above. This graph demonstrates that the magnetization level increases with the laser intensity. Although electron magnetization starts already for relatively low laser intensities, strong proton magnetization also starts to kick in if one goes to high intensity, i.e., $I_L \lambda_L \sim 10^{21}$ W µm$^{-2}$ cm$^{-2}$. As mentioned above, the $B$-field then deflects protons and damages the beam emittance, one of the outstanding features of TNSA protons[1]. We note that since the highest-energy ions are accelerated on axis where the $B$-field vanishes, the impact of the $B$-field upon them is harder to assess. However, the transverse motion of these on-axis ions is unstable, so that they may be rapidly deflected by the $B$-field. Finally, note that the electrons will be effectively less magnetized for ultrashort duration lasers

(i.e., below 100 fs): this is predominantly due to the short plasma expansion during the laser pulse, so that the electrons experience weaker deflections relative to the sheath extent (Supplementary Notes 6 and 7).

**Evidence for magnetization effects at high laser intensity**. The above considerations suggest that self-generated $B$-fields are likely to impact proton acceleration increasingly when raising the laser intensity. To test this, we conducted two experiments (see Methods) geared toward investigating proton acceleration under tight focus conditions in order to maximize the laser intensity, thus falling within the parameter range of Figs 1 and 2. This was achieved by means of re-focusing ellipsoidal plasma mirrors (EPM[31,40], see Methods and the setup shown in the inset of Fig. 3).

Figure 3 summarizes the maximum proton energy recorded in these experiments (see open symbols) as a function of the peak laser intensity over three orders of magnitude, up to $1.3 \times 10^{21}$ W µm$^2$ cm$^{-2}$. The experimental data are fairly well reproduced by 2D PIC simulations (see filled symbols) performed in the same conditions. The solid lines plot the results of the 1D plasma expansion model employed in Figs 1 and 2, which neglects magnetization effects. These analytical predictions reasonably agree with the experimental measurements, performed at LULI (blue symbols) and SNL (red symbols), at low laser intensities. However, both 2D PIC simulations and experimental results gradually deviate from the model's predictions when the laser intensity exceeds $10^{21}$ W µm$^{-2}$ cm$^{-2}$: compare the 1D model-predicted solid red line vs. the experimental data (red open circles) and simulations (red filled circles) for intensities above $10^{21}$ W µm$^2$ cm$^{-2}$.

For laser intensities below $10^{20}$ W µm$^2$ cm$^{-2}$, the consistency between the 1D analytical model, the 2D simulations and the experimental measurements indicates that the acceleration is quasi-1D, as already demonstrated in a number of studies[4,23,41]. For higher laser intensities, the progressive deviation between the 1D unmagnetized model on the one side, and the data and 2D simulations on the other side points to increasing multi-dimensional effects. This is consistent with the increasing electron and ion deflections (highlighted in Figs 1 and 4) induced by $B$-fields of growing strength, which can be only modelled in a multi-dimensional geometry. Under our long-pulse conditions, we

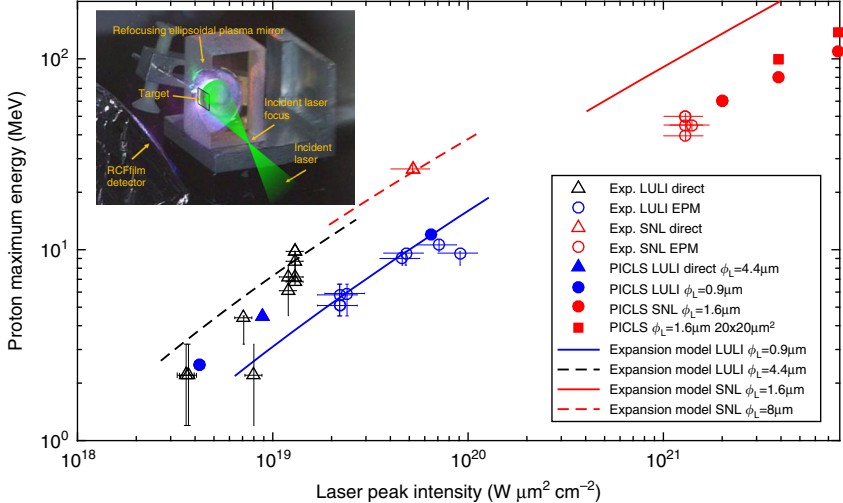

**Fig. 3** Experimental maximum proton energy vs. on-target peak intensity ($I_L\lambda_L^2$). For the LULI experiments (black or blue open symbols), the targets were Al and Au foils of thickness between 0.5 and 2 μm and of transverse dimensions > 1 × 1 mm². For the SNL experiment (red open symbols), the targets were 1.1 μm thick Au foils with transverse dimensions between 50 × 50 μm² and 300 × 300 μm² (note that no trend on the size was observed). Each point corresponds to a single laser shot. Lines plot predictions of the 1D plasma expansion model considered in Figs 1 and 2, and which neglects magnetic field effect (see Methods). Filled symbols represent 2D PICLS simulation results with 2 μm-thick Al targets: filled circles and triangles are for 50 ×50 μm² wide targets and filled red squares for 20 × 20 μm² wide targets. The error bars on the proton energy quantify the energy range of the radiochromic films used to diagnose the TNSA protons. The error bars in the laser intensity arise from the uncertainty in estimating the encircled laser energy within the FWHM spot. (inset) Experimental setup employing a refocusing ellipsoidal plasma mirror (EPM) to reduce the laser focal spot and increase the laser intensity[31]

expect that 2D simulations capture the ion acceleration to relatively good accuracy since the protons should reach their saturation energy within the laser pulse, i.e., during the isothermal acceleration stage[19]. Indeed, as was shown in refs [42,43], differences between 2D and 3D simulations mostly arise during the adiabatic acceleration stage (i.e., after the laser irradiation) for protons that have not already reached their final energy in the isothermal stage.

The perturbations caused to the proton angular distribution by the self-generated $B$-fields at high laser intensities are illustrated in Fig. 4f–h, which display the angular distribution of the proton beam generated at $I_L\lambda_L^2 = 1.3 \times 10^{21}$ W μm²cm⁻² (see also Fig. 4i–k which present azimuthally averaged angular lineout of the proton dose distributions observed in the films shown in Fig. 4f–h). It is characterized by a hollow ring structure with the proton flux peaking at a finite angle with respect to the target-rear normal. We stress that this ring pattern is observed only for high protons energies (compare Fig. 4f with Fig. 4g, h), and disappears when lowering the laser intensity (Fig. 4c–e), in which case the protons exhibit the standard bell-shaped profiles expected for TNSA[1]. 2D PICLS simulations performed under comparable conditions confirm that for the highest laser intensity, the proton beam is deflected outward and hollowed out (Fig. 4a, b). Moreover, both simulation and measurements show similar variations of the angular peak with the proton energy range (as shown in Supplementary Note 3 and Supplementary Fig. 4) The measured angular peak allows us to infer the average field experienced by protons in a specific energy range. The deflection angle is given by $\theta = e\langle B\rangle l/m_p v_{||}$, where $\langle B\rangle$ is the average $B$-field strength, $l$ is the longitudinal extent of the magnetized region and $v_{||}$ is the longitudinal proton velocity. Taking $l \sim 5$ μm as suggested by the simulation of Fig. 1e and a typical deflection angle of 12° as suggested by Fig. 4g, j, we deduce that protons of energy $E_p = m_p v_{||}^2/2 = 25$ MeV undergo an average field $\langle B\rangle \sim 0.3$ GG. This estimate is in reasonable agreement with the results of both the PICLS simulation and the 1D analytical model (Fig. 2a).

The observed ring pattern could be potentially attributed to a different effect, such as hole boring[44]. Since thin (1.1 μm thick) gold foils were used as targets, light pressure could be strong enough to bore through the entire thickness of the foil, and disrupt the laminar shape of the sheath field (note, however, that this scenario could not explain the observed proton energy dependency of the ring diameter). To evaluate the effectiveness of this effect, we can derive the hole boring velocity from the conservation of momentum and energy fluxes across the irradiated region: $v_p = [(1 + R)I_L(t)\cos\alpha/2m_i n_i c]^{1/2}$, where $\alpha = 23°$ is the laser incidence angle on target, $R$ is the laser reflectivity, $m_i = 3.3 \times 10^{-25}$ kg is the ion mass, and $n_i = 5.9 \times 10^{22}$ cm⁻³ is the ion density in the case of gold. For a Gaussian laser pulse of peak intensity $I_L = 1.3 \times 10^{21}$ W cm⁻² and assuming $R = 0.5$[4], the hole-boring depth is estimated to reach 0.8 μm at the peak of the 800 fs duration pulse, while it takes another 300 fs for the laser beam to break out through the target rear. At that moment, however, according to the PICLS simulation, proton acceleration is already completed and exhibits a clear signature of magnetic deflections in the sheath field. Finally, note that another alternative scenario[45], which would invoke resistive $B$-fields to account for similar proton ring structures, would need much thicker targets (>100 μm) to be operative.

## Discussion

We will now discuss several strategies to limit the magnetization effects on proton acceleration. First, an important factor is the relative timescale of the $B$-field growth compared to the proton acceleration timescale (which is of the order of the laser pulse duration). Experimentally, at $I_L\lambda_L^2 = 2 \times 10^{19}$ Wμm²cm⁻², we have measured that the $B$-fields grow to their maximum strength over > 100 fs (Fig. 6 in ref. [29]). In the present investigation, the $B$-fields were hence strongly impacting the electron dynamics since we used relatively long (> 400 fs) laser pulses. Our PIC simulations (Supplementary Fig. 9) and analytical

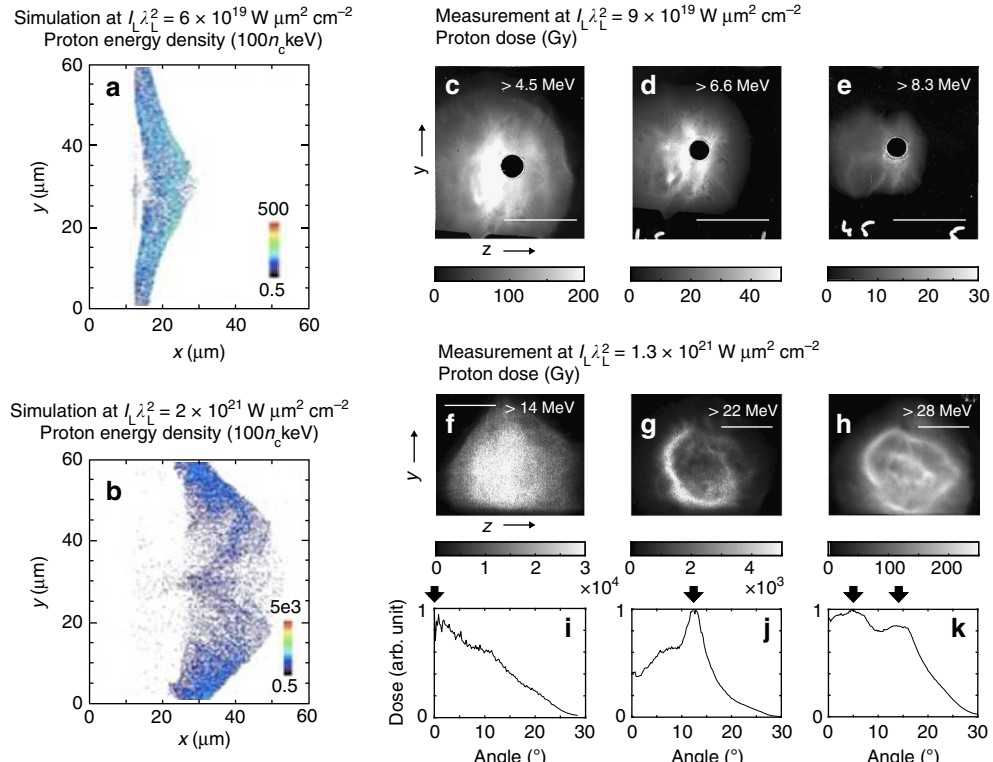

**Fig. 4** Magnetic deflections of protons at high laser intensity. **a, b** Maps of the proton energy density (in units of $100\,n_c$ keV, where $n_c = 1.1 \times 10^{21}$ cm$^{-3}$ is the critical density at 1 μm wavelength) as recorded, at the time of the laser peak, from the 2D PICLS simulations illustrated in Fig. 1a and e, respectively. **b** clearly reveals that at high laser intensity, the protons at the acceleration front are deflected by the magnetic field, forming a ring-like pattern. **c–h** Experimental 2D proton dose distributions (in Gy) measured using stacks of calibrated radiochromic films. In **c–e**, the laser parameters are $I_L\lambda_L^2 = 9 \times 10^{19}$ W μm$^2$ cm$^{-2}$, $\lambda_L = 0.5$ μm, $\phi_L = 0.9$ μm, $\tau_L = 400$ fs, and the target consists of a 0.5 μm thick Al foil. In **f–h**, the laser parameters are $I_L\lambda_L^2 = 1.3 \times 10^{21}$ W μm$^2$ cm$^{-2}$, $\lambda_L = 1$ μm, $\phi_L = 1.5$ μm, $\tau_L = 800$ fs, and the target is a 1.1 μm thick Au foil, yielding a ring-like pattern on the proton dose distribution, consistent with the simulation shown in **b**. For the > 28 MeV protons, this ring pattern encircles what looks like a central jet, which may result from some high-energy protons emitted on axis and having experienced relatively weak deflections. The white bars in **c–h** indicate an angular spread of 20°. **i–k** Proton dose vs. angle with respect to the target-rear normal, as extracted from the proton distribution displayed in **f–h**, respectively. Arrows indicate the angular peaks

calculations (Supplementary Fig. 8) suggest that when using very short (<100 fs) laser pulses, the protons are accelerated quickly enough so that the electron Larmor radius is still larger than the longitudinal sheath extent (i.e., the electrons are weakly magnetized). This would be consistent with the fact that, experimentally, proton acceleration with ultrashort laser pulses seems to display a better scaling vs. laser intensity than with longer pulses (Supplementary Fig. 5). To confirm this, precise in situ measurements of the B-field temporal dynamics (e.g., using a magneto-optical effect in the XUV wavelength range[30]) at high laser intensities and short pulse durations are needed. Nonetheless, we expect that at very high intensities, even for the shortest pulse durations, the magnetic field strength will unavoidably become large enough that the magnetization effect highlighted here will start to be effective, and hence pose a fundamental limit to laser-based ion acceleration.

We tested in the simulations yet another possible strategy for minimizing the B-fields by using small-width targets (so-called reduced mass targets or RMTs), in which the lateral recirculation of electrons homogenizes the sheath and relaxes its transverse gradient of the longitudinal E-field[13]. However, the B-field is also produced by the longitudinal gradient of the transverse field, $\partial E_y / \partial x$, which is enhanced in the case of RMT due to a higher hot-electron density. As a consequence, our simulation shows no noticeable reduction of the magnetic field amplitude with RMT (see Supplementary Note 8 and Supplementary Fig. 10). This

agrees with the trend seen in Fig. 3 (red filled squares): the proton energy is only improved by about 30–40% (mainly as a result of an increased hot electron density and sheath field) when using $20 \times 20$ μm$^2$ wide foils instead of $50 \times 50$ μm$^2$ wide foils. Note that in particular circumstances (e.g., when irradiating the tip of an RMT in the presence of a large scale preplasma[46]), self-generated B-fields can be used to compress the sheath and enhance proton acceleration.

The magnetic inhibition effect highlighted in this study will likely impact not only sheath-accelerated ions, but also alternative ion acceleration schemes, such as relativistic transparency[47] and radiation pressure[12,14,48], that are envisioned to be efficient at even higher laser intensities than discussed here. In principle, radiation pressure acceleration does not require, and even desires to minimize, hot electrons which are at the source of the B-fields investigated here; hence it could be seen to be immune to magnetic inhibition, as supported by simulations of thin targets irradiated by high-contrast, ultraintense laser pulses (Supplementary Fig. 6). However, as recently shown[14], at high laser intensities, tight laser focusing tends anyway to produce hot electrons, which could reintroduce magnetic inhibition of ion acceleration. Undoubtedly, the impact of this previously overlooked process will need to be taken into account when planning for ion acceleration experiments on next-generation ultrahigh intensity laser facilities such as APOLLON, ELI or CALA.

## Methods

**Experiments.** The experiments investigating proton acceleration from solid targets at high laser intensities were performed at the Laboratoire pour l'Utilisation des Lasers Intenses (LULI, France) and Sandia National Laboratory (SNL, NM, USA). Both employed either direct irradiation of the targets positioned at the laser focus, or refocusing of the laser by an ellipsoidal plasma mirror (EPM)[31,40,52]. In the first case, the laser is focused using an $f/2.7$ off-axis-parabolic (OAP) focusing mirror at LULI ($f/4$ at SNL) to a $4.4 \pm 0.5\,\mu m$ ($8 \pm 0.5\,\mu m$ at SNL) full-width at half-maximum (FWHM) spot. This 'direct shot' configuration leads to on-target peak intensities as high as $I_L \lambda_L^2 = 1.3 \times 10^{19}\,W\,\mu m^2 cm^{-2}$ at LULI and $6 \times 10^{19}\,W\,\mu m^2 cm^{-2}$ at SNL. The peak intensity, $I_L$, used in this paper is evaluated by assuming a Gaussian profile for the beam within the FWHM spot size $\phi_L$: $I(r, t) = I_L \exp\left(-4 \ln 2(r/\phi_L)^2 - 4 \ln 2(t/\tau_L)^2\right)$, where $\tau_L$ is the FWHM laser duration (400 fs at LULI and 800 fs at SNL). Using as a practical parameter the laser energy contained in the FWHM spot, $E_L$, one obtains $I_L = 16\left(\frac{\ln 2}{\pi}\right)^{3/2} \frac{E_L}{\phi_L^2 \tau_L} \approx 1.66 \frac{E_L}{\phi_L^2 \tau_L}$. Note that since the laser pulse durations employed here are not extremely short, all of the above intensity calculations do not depend on spatiotemporal couplings within the laser pulse, which can, however, affect the intensity distribution of broadband, ultrashort laser pulses[49].

To generate higher intensities on target, EPMs were placed behind the focus of the laser produced by the OAP (see the inset of Fig. 3 for the setup). The EPM is designed as an ellipsoid of revolution around its major axis, ($x$), $x^2/a^2 + r^2/b^2 = 1$, where $(a,b) = (3.5, 2.012)$ mm at LULI and (12.25, 7.0) mm at SNL, yielding an eccentricity of $\in = 0.818$. The EPM is made of glass treated with an anti-reflection coating at the laser wavelength. It refocuses the laser with a change in the beam numerical aperture. The final FWHM spot size is then reduced to $0.9 \pm 0.1\,\mu m$ at LULI and $1.5 \pm 0.2\,\mu m$ at SNL (as measured by a CCD coupled with a microscope objective of numerical aperture NA = 0.6), yet at the cost of lowering the beam energy. The laser spot at the EPM focus was measured and optimized before each shot. Taking into account the plasma mirror reflectivity and the reduced encircled energy in the focal spot when the plasma mirror is triggered[31], the peak laser intensities are estimated to attain $9 \times 10^{19}\,W\,\mu m^2 cm^{-2}$ at LULI and $1.7 \times 10^{21}\,W\,\mu m^2 cm^{-2}$ at SNL. The laser intensity on the EPM surface was of $3-5 \times 10^{14}\,W\,cm^{-2}$, similar in both experiments and corresponding to a cumulated fluence on the EPM surface of $130-180\,J\,cm^{-2}$ around the peak of the pulse, a standard optimal value for plasma mirrors[50,51]. The expansion velocity of the EPM surface at critical density is estimated to be $c_s \approx \sqrt{Z_{eff} T_e / A m_p}$, with $A$ the atomic mass, $m_p$ the proton mass and an ionization state $Z_{eff}/Z = 0.7$, assuming all the electrons except the $1s$ and $2s$ of Si and O are ionized[50]. The electron temperature $T_e(t)$ depends on the cumulated fluence[50]. The expansion length of the EPM surface is expected to be slightly larger in the SNL experiment due to longer pulse duration: about 0.07 μm around the temporal peak and 0.19 μm at the ending foot of the pulse (800 fs after the peak of the pulse). To evaluate the impact of this expansion on the beam focus after the EPM, we performed ray-trace simulations, assuming that the plasma expands normally to the local EPM surface. With a 0.19 μm expansion of the reflecting surface, the beam size at the EPM focus increased by 0.2 μm, which is smaller than the diffraction-limited spot size (the spot size vanishes for zero expansion, as diffraction effects are neglected in these ray-trace calculations). More information on the ray-trace calculations is provided in Supplementary Note 9, Supplementary Figs. 11, and 12. Also, note that the paraxial approximation starts to fail in the high-NA operating conditions of the EPM. In this case, and for our laser parameters, the peak intensity of the longitudinal laser electric field at the second focus of the ellipsoid reaches ~10% of that of the transverse laser electric field[52].

Similarly to a plasma mirror[50], the EPM also acts as an ultrafast light reflector that is activated only in the rising edge of the laser pulse, hence leading to high-contrast interaction conditions. To obtain similar high-contrast conditions for the direct shots (and thus allow meaningful comparison of their performance), we made use of frequency-doubled pulses at LULI (i.e., operating at a wavelength of $\lambda_L = 0.528\,\mu m$), and of a planar plasma mirror prior to the laser focus at SNL (operating at the fundamental frequency, $\lambda_L = 1.06\,\mu m$). In the latter case, the planar plasma mirror was positioned 20 mm before the OAP focus to be irradiated at the same fluence level as the EPM.

The targets consisted of 0.5–2 μm thick Al and Au foils at LULI and of 1.1 μm thick Au foils at SNL. At LULI, the targets were irradiated at normal incidence ($\alpha = 0°$), while oblique incidence ($\alpha = 23°$) and $p$-polarization were used at SNL. The required target positioning accuracy (approximately 1–2 μm) at the EPM focus was provided by the NA = 0.6 microscope objective, combined with piezoelectric motors. The accelerated protons were detected using radiochromic films[1,2] positioned 20–30 mm away from the target.

**Analytical model.** The analytical 1D model used in Figs 1–3 is based on the theoretical model developed in ref. [19], which describes the isothermal and collisionless expansion (along the longitudinal $x$ axis in our case, i.e., normal to the target surface) of a plasma into the vacuum, as driven by a population of initial hot electrons' temperature (or energy) $T_0$ and density at the target surface $n_{h,rear}$. The plasma expands into the vacuum due to electrons pulling out protons through the space-charge electrostatic field. The hot electrons are generated through the $\mathbf{J} \times \mathbf{B}$[1] and Brunel[53] mechanisms, with kinetic energies of several MeV for the laser intensities considered here[54,55]. In the present paper, the initial hot-electron temperature was estimated as $k_B T_0 = m_e c^2(\gamma_0 - 1)$, where $\gamma_0 = \pi/2K(-a_0^2)$ is the mean hot-electron relativistic factor, $a_0 = \left(I_L \lambda_L^2/1.37 \times 10^{18}\,W\,\mu m^2\,cm^{-2}\right)^{1/2}$ is the normalized laser field, and $K$ is the elliptical integral of the first kind. This scaling, suggested by Kluge et al.[55], is specifically adapted to the interaction of intense lasers with steep-gradient plasmas, i.e., as is achieved under our high-contrast experimental conditions. Furthermore, we found that this scaling gives the closest match to our numerical simulation results. The hot-electron density at the laser-interaction surface is estimated to be $n_h \approx 0.5\gamma_h n_c$, where $n_c = \varepsilon_0 m_e \omega_L^2/e^2$ is the critical density and $\gamma_h = a_0/\sqrt{2} + 1$ is the relativistic factor derived from one-dimensional energy and momentum flux conservation. Here the factor 0.5 is introduced to take into account the time-averaged value of $\gamma_h$. This gives a total absorbed energy into hot electrons of $E_{h,tot} = N_{h,tot} k_B T_0$, where $N_{h,tot} = \pi n_h \phi_L^2 \tau_L v_e/4$ assuming $v_e \sim c$. This results in a laser-to-hot-electron coupling efficiency $E_{h,tot}/E_L \sim 25 - 40\%$, in the intensity range discussed in this paper. The density at the target rear is $n_{h,rear} = n_h \left(1 + d r_L^{-1} \tan\theta\right)^{-2}$, where the target thickness is $d = 2\,\mu m$, $r_L = \phi_L/2$ is the laser spot radius on target, and the half-angle electron divergence within the target is $\theta \sim 45°$[4]. In the model, the initial (maximum) electric field in the sheath is given by $E_0 = \sqrt{n_{h,rear} k_B T_0/\varepsilon_0}$. We extracted the proton energy at $t = \tau_L$. At that moment, according to the PICLS simulation, proton acceleration is already completed for the case of laser pulse durations we explored here.

The azimuthal magnetic field at the target rear is assumed to be generated by the time-dependent Faraday law, $\partial B_z/\partial t = \partial E_x/\partial y - \partial E_y/\partial x \sim \partial E_x/\partial y$, assuming a 2D geometry with $\mathbf{B} = (0,0,B_z)$. It leads to steady growth of the magnetic field up to the stage when the magnetic pressure becomes comparable with the plasma pressure. We note that other sources (e.g., gradients of density and temperature[29]) may induce magnetostatic fields on the target surfaces, yet these processes occur over time-scales longer than the ion acceleration time scales of interest here. In this frame, assuming a Gaussian transverse profile for the longitudinal sheath field, $E_x(y) = E_0 f(t)e^{-(y/r_y)}$, we can approximate its peak transverse gradient as $(\partial E_x/\partial y)_{max} \sim E_0 f(t)/r_y$. The inductive $B$-field is then estimated to be $B_z(t) = (E_0/r_y) \int_0^t f(t')d\tau'$. We consider here the electric field in the plateau region of the sheath where most of the particles are confined. During the isothermal phase ($0 \leq t \leq \tau_L$), we have $f(t) = 2/\sqrt{2e_N + \omega_{pi}^2 t^2}$, with $\omega_{pi} = \sqrt{n_{h,rear} e^2/m_p \varepsilon_0}$, which yields $B_z(t) = (E_0/r_y \omega_{pi})\ln\left|a + \sqrt{1 + a^2}\right|$, where $a = \omega_{pi}t/\sqrt{2e_N}$. The velocity of the accelerating protons increases a $v_p(t) = 2c_{s0}\ln\left|a(t) + \sqrt{1 + a(t)^2}\right|$, and the position of the ion front is then given by $x_{front}(t) = \int_0^t v_p(\tau)d\tau$. After the laser pulse, as the electrons progressively give their energy to the ions and cool down in the expansion, $T_e$ decreases with time (Fig. 1b). Last, the electron and proton Larmor radii are calculated using $R_L^p(t) = m_p v_p(t)/eB_z(t)$ and $R_L^e(t) = m_e c\sqrt{\gamma(t)^2 - 1}/eB_z(t)$, where $\gamma(t) = 1 + k_B T_e(t)/m_e c^2$.

**Numerical simulations.** To analyze the plasma dynamics at play during and following the intense laser irradiation, we resort to PIC numerical codes, which provide a first-principles simulation framework more adequate than magneto-hydrodynamics (MHD) codes. Indeed, MHD is based on the assumptions of quasi-neutrality, small particle Larmor radii, and thermal particle distributions. While part of these assumptions may locally hold off-axis and inside the expanding electron–ion plasma (where quasi-neutrality holds, the $B$-field is at its strongest, and the electrons may gyrate with radii smaller than the plasma/field scale-lengths), MHD is invalid at the ion front (where the quasi-neutrality assumption breaks down) and/or around the axis (where the $B$-field weakens and/or changes sign), that is, in the regions where the maximum ion energies are to be found.

The 2D PIC numerical simulations presented in this work are performed using the PICLS, PICADOR, and CALDER codes. PICLS features binary collisions among charged particles and dynamic ionization[26]. Absorbing boundary conditions are used for particles in the transverse direction (i.e., no electron reflux is imposed to represent the actual large transverse size of the target). The target consists of a neutral plasma of electrons and $Al^{3+}$ ions. A 20 nm thick layer of protons is added at the target rear surface to mimic the surface contaminants. The resulting two-layer target can lead to modulations in the low-energy side of the proton spectrum[56], especially if the protons in the contaminant layer are depleted[57,58], but this cannot affect the proton maximum energy[56], the observable on which we concentrate here. The ion density is initialized to $5 \times 10^{22}\,cm^{-3}$, while the electron and ion temperatures are both set to zero. The electron density increases dynamically during the laser irradiation via ionization processes. The spatial (resp. temporal) resolution was 1/50 of the wavelength (resp. laser oscillation period). The laser focal spot and temporal shape obey Gaussian distributions. The collisionless PICADOR code is run on heterogeneous cluster systems including Xeon Phi coprocessors[59]. The PICADOR simulations use exactly the same parameters as the PICLS simulations, except that they neglect the ionization dynamics (i.e., the ion charge state keeps a constant value, $Z = 3$, during the simulations). The CALDER[60] simulations discussed in Supplementary Note 5 consider fully ionized carbon nanometric foils assuming negligible Coulomb collisions.

**Code availability**. The Matlab code solving the 1D model used in Figs 1–3 is available from the corresponding authors upon reasonable request.

**Data availability**. The data that support the findings of this study are available from the corresponding authors upon reasonable request.

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

## Acknowledgements

We acknowledge the expert support of the ELFIE and Sandia 100 TW laser operation crews and engineering teams. We thank Y. Kuramitsu and G. Revet for useful comments and discussions. This work was supported by grant E1127 from Région Ile-de-France, by

CREST Japan Science and Technology, and in part by the Ministry of Education and Science of the Russian Federation under Contract No.14.Z50.31.0007. This project has received funding from the European Union's Horizon 2020 research and innovation programme under grant agreement no 654148 Laserlab-Europe, and was partly done within the LABEX Plas@Par project and supported by Grant No. 11-IDEX-0004–02 and ANR-17-CE30–0026 PiNNaCLE grant from Agence Nationale de la Recherche (France). The simulations were partially performed on resources provided by the Joint Super-computer Center of the Russian Academy of Sciences. A. Kon acknowledges support from the JSPS Global COE program. M.N. was partially supported by a JSPS Postdoctoral Fellowship for Research Abroad. Y.S. was supported by the JSPS KAKENHI under Grant No. JP15K21767 and the DOE-OFES under Grant No. DE-SC0008827. L.H. acknowl-edges support by the NASA Earth and Space Science Fellowship Program under grant NNX13AM28H. Sandia National Laboratories is a multi-mission laboratory managed and operated by National Technology and Engineering Solutions of Sandia, LLC., a wholly owned subsidiary of Honeywell International, Inc., for the U.S. Department of Energy's National Nuclear Security Administration under contract DE-NA0003525. Support was also provided by Sandia's Laboratory Directed Research and Development program. The CALDER simulations were performed using HPC resources at TGCC/CCRT (Grant No. 2013–052707).

## Author contributions

M.N., R.K., and J.F. designed the project, M.N., S.B., A.Kon, M.G., L.H., P.R., M. Sc and J.F. performed the experiment, M.N., J.F., S.B., and A. Kon analyzed the data, Y.S., A. Kor., and L.G. performed the simulations, B.A., P.A.M.K., J.S., M. St., and R.K. supported the students and the project, M.N., Y.S., A. Kor., S.N.C., L.G. and J.F. wrote the paper. All authors discussed the results and reviewed the paper at its various stages.

## Additional information

**Competing interests:** The authors declare no competing financial interests.

