## [Peer Review File · Nature Communications]

Reviewers' comments:

Reviewer #1 (Remarks to the Author):

The manuscript "On magnetic inhibition of laser-driven, sheath accelerated high-energy protons," submitted by N. Nakatsumi, et al., present experimental and numerical simulations of ion acceleration from overdense laser plasma interactions. In particular, they provide evidence of a magnetically induced limit for maximum proton energy. If true, it has important implications for the future of compact ion accelerators and solves a long standing issue of why high intensity systems (which incidentally use tighter and tighter focii, which the authors show exacerbates the magnetic field generation) do not produce the high ion energies initially hoped for. There are a few points which I would like more explanation on, but if explained the manuscript seems like an important topic for planning and advancing future laser based ion accelerators. More comments follow.

-When reducing the target transverse size, the magnetic field strength is supposed to be pretty limited, however the gains are only about 30-40% (and the authors note that this is also due to the higher electron density). Shouldn't the results in this case match the expansion model? It seems that the magnetic field effect is a perturbation compared to the physics that drops the energy down by an order of magnitude when compared to the lower intensities.

-There is no mention of the 3D effects which might contribute. The authors do a good job showing how their results match up well with lower dimensional models, but it would help to add a small conversation on it.

-The 2D PIC simulations don't mention protons, which are the ion of interest in this experiment. Also, it is well known that for higher intensities two layer effects can take place, influencing the interaction. There does not seem to be any reason why not to include a thin proton layer on the back. I am guessing that this is an accidental omission, since proton data from the simulations exist in the SI.

-Fig S1c may have the wrong units reported in the caption. It also might be more useful as a figure to show a tighter color bar that more closely matches the proton density, since the sheath electrons are what matters and they are completely washed out in this figure.

-The experimental data showing that the high energy electrons are behaving due to different physics is a nice piece of experimental data. Considering the prior research the authors have done on the subject, it is surprising that no other experimental diagnostic or data exist to complement the data. Is there no electron data or anything that helps make the case?

Reviewer #2 (Remarks to the Author):

The paper by Nakatsumi et al. focuses on a particular aspect of target normal sheath acceleration (TNSA) of ions, namely on magnetic fields, self-generated on a target surface, irradiated by a highly intense laser pulse. The authors claim that these fields might play detrimental role on ion acceleration, degrading their quality and hampering their maximum energy. In my opinion, the claim has some novelty in it, in particular because a case of tight focusing is explored. I find the explanation of underlying physics clear and easy to follow, but have some issues with scientific soundness of the experimental and simulated results.

1. I have a question regarding experimental intensity calculation when ellipsoidal plasma mirror was

used. As far as I understood from the methods section, it was estimated assuming certain pulse durations (400 fs for LULI and 800 fs for SNL), focal spot sizes, and energies contained within FWHM of the focal spots. That would be correct if longitudinal extent of the laser pulse were less than its Rayleigh length. But that is not the case since the laser pulse is quite long, while its Rayleigh length is, on the opposite, very short. The pulse "does not fit" within its Rayleigh length. Because of that, laser pulse intensity is effectively much smaller than what is expected based on its usual definition [Jeong et al., *Opt. Express* 23, 11641-11656 (2015); Pariente et al., *Nature Photonics* 10, 547-553 (2016)]. If this is true, then decreased maximal energy of accelerated protons might be explained by lower (than estimated) effective laser pulse intensity on target.

2. All simulations performed in the paper are 2D ones. It was already shown that 2D and 3D simulations might give significantly different results, with 3D ones demonstrating weaker TNSA, resulting in smaller maximal energies of accelerated protons [Novo et al., *Sci. Rep.* 6, 29402 (2016); Sgattoni et al., *Phys. Rev. E* 85, 036405 (2012)]. The authors should provide clear evidences of possibility to use 2D simulations for their work. For example, a comparison of results obtained with 2D and 3D simulations for a particular set of experimental conditions.

3. The authors claim that the self-generated magnetic fields pose fundamental limit for TNSA. In fact, it is not a fundamental one, but instead is a consequence of a particular set of experimental conditions: tight focusing of a long laser pulse. As the authors discuss later in the paper, deleterious effects can be decreased by using short laser pulses and loose focusing.

4. Figure 2 presents results of simulations and one experimental datapoint, but the datapoint is taken from an experiment with loose focusing, while the simulations were performed with tight focusing. As it is shown by the authors, those two situations result in quite different outcomes, so I don't see how they can be put on one plot. Also, the datapoint should have an associated errorbar.

Given these questions regarding scientific soundness of both experimental and simulated results, I cannot recommend the paper for publication in Nature Communications.

Reviewer #3 (Remarks to the Author):

The authors claim that the B-field generation at the target rear surface hamper the TNSA ion energy. To highlight their idea, the authors provide 2D PIC simulations and a couple of elementary formulas like calculating particle cyclotron radius and rough estimates for the B-field.

The only evidence they provide that the B-field influences the TNSA mechanism (of course it does - one has to solve corresponding MHD equations) - they compare 1D TNSA model with 2D PIC simulations. The difference falls fairly into the statistical scatter. For example, "Exp. SNL direct" data in Fig. 3 is factor 2 above the analytical trend, while "SNL EPM" data are factor 2 below the trend. In general, the influence of B-field on both electron dynamics and ion acceleration are doubtless. Still, I am not convinced that the paper in this form satisfies all Nature conditions. Maybe, it should be published in a more specialized journal.

Answers to reviewers comments, paper NCOMMS-16-19781-T, Nakatsutsumi et al.

In the following, comments by the referees are numbered and in *italic*. Our responses are just below.

Modifications to the main text in response to the reviewers' comments and to improve the clarity of the text are shown **highlighted in yellow** in the manuscript.

We hope that we met the expectations of the reviewers, as well as yours, and that our article could be considered for publication.

The authors

Reviewer #1 (Remarks to the Author):

The manuscript “On magnetic inhibition of laser-driven, sheath accelerated high-energy protons,” submitted by N. Nakatsumi, et al., present experimental and numerical simulations of ion acceleration from overdense laser plasma interactions. In particular, they provide evidence of a magnetically induced limit for maximum proton energy. If true, it has important implications for the future of compact ion accelerators and solves a long standing issue of why high intensity systems (which incidentally use tighter and tighter focii, which the authors show exacerbates the magnetic field generation) do not produce the high ion energies initially hoped for. There are a few points which I would like more explanation on, but if explained the manuscript seems like an important topic for planning and advancing future laser based ion accelerators. More comments follow.

1-When reducing the target transverse size, the magnetic field strength is supposed to be pretty limited, however the gains are only about 30-40% (and the authors note that this is also due to the higher electron density). Shouldn't the results in this case match the expansion model? It seems that the magnetic field effect is a perturbation compared to the physics that drops the energy down by an order of magnitude when compared to the lower intensities.

As the reviewer correctly points out, reducing the transverse size of the target (i.e. using so-called “reduced mass target”, or RMT) increases the proton energy due to the lateral confinement of the refluxing electrons, which leads to an increase in the effective density of the electrons within the sheath [^{*}]. We have also pointed out, in our previous version of the manuscript, that in RMTs, the electron sheath is flatter, which reduces the transverse gradient of the sheath electric field $\partial E_x / \partial y$. This would seem to lead to a lower B-field, as pointed out by the reviewer, and to a mitigation of the magnetic inhibition of proton acceleration. However, this is not the case as **in RMT the B-field remains quite significant** (similar in strength as to the one generated in a large size target), and thus the magnetic inhibition effect too, which explains the fact that proton energies do not match the energies predicted by the unmagnetized (1D) model.

To detail and clarify this, we run new simulations, which are presented below (Fig.A) and reported also in the *Suppl. Info.* (Section 8). For these simulations, the parameters of the laser pulse and target were kept equal to those used in Fig. 1b of the main manuscript (see caption in Fig.A), except that the target width was reduced to 20 μm . These simulations were performed using the PICADOR code (see Methods for details).

Fig. A (d) shows that the transverse gradients of the magnetic field on the rear side of the target are indeed reduced as to when using a large width target (as in Fig.1 of the main paper). However, it appears that **the RMT B-field is comparable in strength and topology to that in the case of a large target**. This is due to the fact that the B-field is also (and mainly in this case) produced via the $\partial E_y / \partial x$ term (which is connected to the transverse currents on the

^{*} The increase in the electron sheath density in RMTs was discussed in detail in our previous paper [PRL **105**, 015005 (2010)] as well as in other papers [Applied Physics Letters **103**, 054102 (2013), Plasma Phys. Control. Fusion **56** 084004 (2014), Phys. of Plasmas **22**, 043116 (2015)].

target surface). This term is strong in the case of the RMT due to higher densities of electrons. We also note that in general it is not surprising that the B-field strength would be similar in the case of a large target or a RMT since the source of the magnetostatic field is an electric current formed by electrons expelled by the laser pulse (see Section 8 of the *Suppl. Info.* For more details). Since such current depends only on the number of hot electrons injected by laser pulse and their velocities, and thus mainly depend on the laser intensity and target density, it can therefore be expected, as revealed by comparing the B-field in Fig.1 of the main text and in Fig.A, that the generated magnetic fields are comparable for both the normal target and RMT cases.

This magnetic field has enough strength to magnetize electrons as can be seen from the electron trajectories shown in Fig. A (c). As a consequence, **the strong B-field still acts detrimentally on proton acceleration in RMT** (through the same mechanisms as described in the manuscript). This explains why we see in Fig. 3 of the main paper only a partial increase of proton energies in the case of RMTs compared to the case of normal targets. And it explains why, as the reviewer points out, in our case at very high intensities ($> 10^{20}$ W/cm²), that even with RMT we do not go back to the level of proton energy predicted by the simple 1D, unmagnetized model (which doesn't take into account the magnetic field).

These points are added in the *Supplementary Information*, and are now clarified in the revised version of the paper, as follows (the modified text is highlighted in yellow) on p. 13:

“We tested in the simulations yet another possible strategy for minimizing the B-fields by using small-width targets (so-called reduced mass target or RMT), in which the lateral recirculation of electrons homogenizes the sheath and relaxes its transverse gradient of the longitudinal E-field [13]. However, the B-field is also produced by the longitudinal gradient of the transverse field; dE_y/dx , which is enhanced in the case of RMT due to a higher densities of electrons. As a consequence, our simulation shows no noticeable reduction of the magnetic field amplitude with RMT (details in the Section 8 of the *Supplementary Information*). This is seen in Fig. 3, with red closed squares: the proton energy is only improved by about 30 – 40 % (due predominantly to an increased hot electron density and sheath field) when using $20 \times 20 \mu\text{m}^2$ wide foils compared to when using $50 \times 50 \mu\text{m}^2$ wide foils.”

We should also note that a very recent paper [Physics of Plasmas 23, 053114 (2016)] discussed a strategy employing RMT surrounded by a very large preplasma in order to exploit B-fields to enhance proton acceleration (through a compression of the sheath). Here the physics is mostly linked with previously observed field-enhancement at the tip of a wire/RMT [Phys. Rev. Lett. 110, 215004 (2013), Appl. Sci. 5, 459-471 (2015)]. The effect highlighted in those papers is not at play in our case since we use high-contrast laser pulses and do not exploit the tip/edge of our targets.

We nonetheless briefly mention it in the revised version of the paper on p.13-14:

“Note that in particular circumstances (when irradiating the tip of an RMT in the presence of a large scale preplasma [48]), self-generated B-fields can be used to compress the sheath and enhance proton acceleration.

Figure A: 2D PIC simulations (using the PICADOR code) of proton acceleration in a RMT target. The parameters of the laser pulse and target were kept equal to those used in Fig. 1b of the main manuscript (the laser FWHM spot size ϕ_L , FWHM duration τ_L , wavelength λ_L and intensity $I_L \lambda_L^2$ are, respectively, $1.6 \mu\text{m}$, 700 fs , $1 \mu\text{m}$ and $2 \times 10^{21} \text{ W} \cdot \mu\text{m}^2 \cdot \text{cm}^{-2}$, with $2\text{-}\mu\text{m}$ thick solid aluminium target), except that the target width was reduced to $20 \mu\text{m}$. Shown are (a) electron and (b) ion concentrations, (c) trajectories of sampled energetic electrons and (d) the transverse B-field (B_z) distribution. All snapshots are taken at the peak of the laser pulse.

2-There is no mention of the 3D effects which might contribute. The authors do a good job showing how their results match up well with lower dimensional models, but it would help to add a small conversation on it.

This discussion was indeed lacking from our initial manuscript, although it first was through a 3D simulation (in the seminal paper of Pukhov [Phys. Rev. Lett. **86**, 3562 (2001)]) that our attention was attracted to the possibility of the detrimental effect played by the B-fields.

There have been, that we are aware of, few papers discussing proton acceleration comparatively in 2D and 3D:

- Sgattoni et al., Phys. Rev. E **85**, 036405 (2012) find that TNSA proton acceleration in 2D and 3D simulations exhibit the same dynamics, but that proton energies in 3D are $\sim 1/2$ lower than in 2D. These 2D/3D comparisons are made at laser intensities of 1.38×10^{20} W/cm².
- d'Humières et al., Phys. Plasmas **20**, 023103 (2013) reach a similar conclusion: 2D simulation give TNSA proton energies that are ~ 1.5 higher than in 3D. All these 2D/3D comparisons are made at laser intensities of 2×10^{20} W/cm².

A very important point relative to these two papers is that their simulations are conducted using very short pulse lasers (25-45 fs). In this case, **the protons reach the saturation of their energy only after the laser pulse has stopped**. This is clearly seen e.g. in Figure B below. This means that the protons experience two phases during their acceleration: (1) an isothermal phase when the laser is on, followed by (2) an adiabatic phase when the lasers is stopped. **Is it precisely when the acceleration regime changes from isothermal to adiabatic that the difference between 2D and 3D appears**, as clearly appears in Figure B below. This difference is due to the fact that there is a quicker adiabatic cooling of the sheath electrons in 3D than in 2D (and since the adiabatic cooling governs the dilution of the accelerating sheath field). In Figure B, the deviation in proton maximum energy between 2D and 3D clearly starts when the (45 fs duration) pulse has stopped.

Figure B: Fig. 6 of the paper by d’Humières et al. [Phys. Plasmas **20**, 023103 (2013)] showing that the deviation between 2D and 3D computed proton energies takes place, when using a 45 fs duration laser pulse in the simulation, only after the laser pulse has stopped. The deviation is obviously due to a difference in the adiabatic phase of the expansion which is here important because the protons have not yet reached their energy at saturation. This is not the case in our paper since the laser pulses we use are long (700 fs for the SNL experiment), hence the protons are fully accelerated during the laser pulse, and hence no deviation between 2D and 3D is expected.

In our case, the pulse duration used in the SNL experiment (performed at the highest intensity used in our study) is 800 fs. This is much longer than the pulse duration used in the above 2 papers. Our PICLS-2D simulation shows that the proton energy in our case saturates around 300 fs after the peak, which is still before the laser pulse ends. **This means that the protons in our experiment acquire their energy before the adiabatic cooling phase takes place;** the sheath field at the centre is maintained thanks to a continuous supply of fresh electrons. This is the reason why our 2D PIC simulations at high intensity fit well with our experimental data, and that **we do not expect that 3D effects would induce a deviation from what is predicted in 2D.**

This said, doing real 3D PIC simulations in our conditions (long pulse duration and fairly large dimensions) is unfortunately still beyond our capability, but we believe that the argument detailed above and based on published papers is solid. As discussed in the manuscript, the deviation of the proton energy from the 1D model is due to the magnetic field. The 1D model intrinsically neglects the magnetic field and thus predicts unrealistically high proton energy at high-intensity.

This is now clarified as follows in the paper on p.10:

“For laser intensities below $10^{20} \text{ W.}\mu\text{m}^2.\text{cm}^{-2}$, the consistency between the 1D analytical model, the 2D simulations and the experimental data indicates that the acceleration is quasi-1D, as already demonstrated in a number of studies [4,42,43]. For higher laser intensities, the progressive deviation between the 1D unmagnetized model on the one side, and the data and

2D simulations on the other side points to increasing multi-dimensional effects. This is consistent with the increasing deflections of the electrons and ions (highlighted in Fig.1 and Fig. 4) induced by B-fields of growing strength, which can be only modelled in a multidimensional geometry. Under our long-pulse conditions, we expect that 2D simulations capture the ion acceleration to relatively good accuracy since the protons should reach their saturation energy within the laser pulse, i.e. during the isothermal acceleration stage [19]. Indeed, as was shown in Refs [44,45], differences between 2D and 3D simulations mostly arise during the adiabatic acceleration stage (i.e. after the laser irradiation) for protons that have not already reached their final energy in the isothermal stage.”

3-The 2D PIC simulations don't mention protons, which are the ion of interest in this experiment. Also, it is well known that for higher intensities two layer effects can take place, influencing the interaction. There does not seem to be any reason why not to include a thin proton layer on the back. I am guessing that this is an accidental omission, since proton data from the simulations exist in the SI.

We apologize for the lack of clarity in the paper about the setup of the simulations – the reviewer is right: we forgot the mention in the *Methods* section that **there is indeed a layer of protons at the target rear surface, which is 20 nm-thick**. It is from this layer, mimicking the Langmuir contaminant layer on solids, that protons are accelerated in the simulations. These protons can be seen in Fig 4.a-b. This is clarified in the *Methods* section (on p.19) as follows:

“The target consists of a neutral plasma of electrons and Al^{3+} ions. **A 20 nm thick layer of protons is added at the target rear surface to mimic the surface contaminants.**”

Regarding the effect of having a double layer structure on the target (a contaminant, proton-rich layer on top of a heavier substrate), there have been many studies on this topic [J. Badziak et al., Phys. Rev. Lett. **87**, 215001 (2001), T.Z. Esirkepov et al., Phys. Rev. Lett. **89**, 175003 (2002); B. Hegelich et al., Nature **439**, 441-444 (2006), H. Schworer et al., Nature **439**, 445-448 (2006), S. S. Bulanov et al., Phys. Rev. E **78**, 026412 (2008), A. Brantov et al., Phys. Plasmas **16**, 043107 (2009) to name a few]. Basically, these studies showed that spectral modulations toward the low energy side of the proton spectrum can develop, but this cannot affect the proton maximum energy.

This is now clarified as follows:

“The fact that we have a double layer system can lead to spectral modulations toward the low energy side of the proton spectrum [58], especially if the protons in the contaminant layer are depleted [59,60], but this cannot affect the proton maximum energy [58], the observable on which we concentrate here.”

4-Fig S1c may have the wrong units reported in the caption. It also might be more useful as a figure to show a tighter color bar that more closely matches the proton density, since the sheath electrons are what matters and they are completely washed out in this figure.

We have double checked the units and believe that they are right. Regarding the color bar, we have checked and do not have the data anymore of these simulations. Since it would be costly to re-run them, and that we preferred using our computer allocation to run new simulations

(see Fig S9 and S10 of the *Suppl. Info.*), we hope the referee will understand that we will leave them as is.

5-The experimental data showing that the high energy electrons are behaving due to different physics is a nice piece of experimental data. Considering the prior research the authors have done on the subject, it is surprising that no other experimental diagnostic or data exist to complement the data. Is there no electron data or anything that helps make the case?

We agree with the referee that angular- and energy-resolved electron data (such as magnetic electron spectrometers in different axes, for example) would have been a good complement to our proton data and to their interpretation. Because the very compact setup required by the use of the ellipsoidal plasma mirror, with a close-by high N.A. imaging system being needed and covering most of space around the interaction target, there were strong constraints on the space available around the target for diagnostics other than an RCF pack to analyze the accelerated protons. This was different from our other experiments which employed standard laser focusing parabola, in which case the whole horizontal plane was available for diagnostics (except the solid angle occupied by the incoming and focusing laser beam).

Now, this is not an excuse not to field more diagnostics, but just to explain the constraints we had during these first test experiments. The next step would be to measure the B-field directly, in situ, using for example, a magneto-optical effect employing a XUV (high harmonics) beam [M. Tatarakis et al. *Nature* 415, 280 (2002)] or X-ray beam. Indeed, using proton [B. Albertazzi et al., *Phys. Plasmas* 22, 123108 (2015)] or electron [W. Schumaker et al., *PRL* 110, 015003 (2013)] radiography would be harder since it would require aligning two targets with a straight line of sight. It would also require an additional high-intensity laser beam (which we didn't have in the SNL experiment), or an external femtosecond light source such as an FEL. This is briefly discussed in the "discussion" section of the main text, on p.13.

It now reads:

"To confirm this, precise **in situ** measurements of the B-field temporal dynamics, **e.g. using a magneto-optical effect employing a XUV (high harmonics) beam [30]**, at very high laser intensity are needed."

Reviewer #2 (Remarks to the Author):

The paper by Nakatsutsumi et al. focuses on a particular aspect of target normal sheath acceleration (TNSA) of ions, namely on magnetic fields, self-generated on a target surface, irradiated by a highly intense laser pulse. The authors claim that these fields might play detrimental role on ion acceleration, degrading their quality and hampering their maximum energy. In my opinion, the claim has some novelty in it, in particular because a case of tight focusing is explored. I find the explanation of underlying physics clear and easy to follow, but have some issues with scientific soundness of the experimental and simulated results.

1. I have a question regarding experimental intensity calculation when ellipsoidal plasma mirror was used. As far as I understood from the methods section, it was estimated assuming certain pulse durations (400 fs for LULI and 800 fs for SNL), focal spot sizes, and energies contained within FWHM of the focal spots. That would be correct if longitudinal extend of the laser pulse were less than its Rayleigh length. But that is not the case since the laser pulse is quite long, while its Rayleigh length is, on the opposite, very short. The pulse “does not fit” within its Rayleigh length. Because of that, laser pulse intensity is effectively much smaller than what is expected based on its usual definition [Jeong et al., Opt. Express 23, 11641-11656 (2015); Pariente et al., Nature Photonics 10, 547–553 (2016)]. If this is true, than decreased maximal energy of accelerated protons might be explained by lower (than estimated) effective laser pulse intensity on target.

The referee touches the fundamental issue of correctly estimating the laser intensity, which is becoming a more and more important issue for the ultrashort, high-intensity laser community.

To gauge the issue of the Rayleigh length in our experiments, we can estimate it using $z_R = \pi \omega_L^2 / \lambda_L \propto f^2 \lambda$ (which is strictly correct only for monochromatic beam, a reasonable assumption in our case, see below). Thus, we estimate our Rayleigh length to be $\pm 3.6 \mu\text{m}$ (LULI exp.) and $\pm 5.8 \mu\text{m}$ (SNL exp) around the focus, respectively. The z_R is indeed much shorter than our pulse duration, as the reviewer pointed out.

Now, as mentioned by the referee, Pariente *et al*, argue that the calculation of the intensity only via laser energy, duration and surface may potentially overestimate the intensity. This would be due to neglecting the phase term of the electric field, and assuming a perfect wavefront of a monochromatic beam. The accurate and rigorous estimate of the light intensity requires the information about spatio-temporal couplings (STCs), or the frequency dependence of the spatial properties of a light beam.

However, it is important to note that the spatio-temporal coupling plays an important role only for broad bandwidth lasers, as the behavior of different frequency components and the coherent superposition between them needs to be taken into account. In our case however, the spectral bandwidth was very narrow (a 2 nm FWHM bandwidth is necessary for producing a 480 fs Gaussian pulse while a 63 nm FWHM bandwidth is required for a 15 fs FWHM pulse). Thus, we can readily assess that **the STCs play a minor role in our case** and clearly do not impact the (usual) calculation of the intensity, as stated in the *Methods* section. To take a concrete example, the one of a long pulse laser, e.g. a 1 ns pulse, such pulse corresponds to distributing the energy over 30 cm in length. This is much longer than the Rayleigh length (z_R) of a standard long-focal-length focusing (usually $z_R \sim$ a few mm). Since the STCs are

also negligible for such pulse, the intensity can indeed be calculated by the power divided by surface over which it is focused. In short, in the case of negligible STCs, the flux on the sample at the ‘focus’ point is irrelevant of the overall Rayleigh length, since, at the focus, the local fields constructively interfere.

Let us also discuss the paper Jeong et al., Opt. Express 23, 11641-11656 (2015). There, the authors claim a shortening of the effective pulse duration for low f-number beams, because of the field intensity quickly decreasing outside the Rayleigh range (see Fig. 7(b) and Fig. 8 (c) of the paper). But this doesn’t mean that there is no energy outside the Rayleigh range, it just means that there is destructive interference of the coherent fields that only add up at focus. The field which was invisible outside the Rayleigh range at a given time t_0 will gradually interfere constructively as time goes on, and generate a high intensity spot at the focus at a later time t_1 . But the duration over which there is a high intensity condition at focus is indeed the “nominal” pulse duration. The time duration of the laser pulse effective irradiation on the target is in fact not related to the Rayleigh length.

To conclude, we believe that our estimation of the focused intensity for our narrow bandwidth laser is justified, within the error bars provided in Fig. 3. For us, having different spectral phases over space and time is not as critical as for ultra-short broadband femtosecond lasers.

Triggered by this important comment by the referee, we added the following sentence in the *Methods* section, on p. 16:

“Note that, since the laser pulse durations employed here are not extremely short, all of the above intensity calculations do not depend on spatio-temporal couplings within the laser pulse, which can however affect the intensity distribution of broadband, ultrashort laser pulses [51].”

2. All simulations performed in the paper are 2D ones. It was already shown that 2D and 3D simulations might give significantly different results, with 3D ones demonstrating weaker TNSA, resulting in smaller maximal energies of accelerated protons [Novo et al., Sci. Rep. 6, 29402 (2016); Sgattoni et al., Phys. Rev. E 85, 036405 (2012)]. The authors should provide clear evidences of possibility to use 2D simulations for their work. For example, a comparison of results obtained with 2D and 3D simulations for a particular set of experimental conditions.

The referee is right that, under certain circumstances (detailed below), there can be significant differences in the maximum proton energies predicted by 2D and 3D simulations. However, as also detailed below, in our case of long pulses (800 fs for the SNL experiment, the one conducted at high intensity) we believe that the 2D simulations are relevant and that 3D effects do not significantly deviate from the 2D prediction, notably as regards the time at which magnetic inhibition sets in.

There have been, that we are aware of, few papers discussing proton acceleration comparatively in 2D and 3D:

- Sgattoni et al., Phys. Rev. E **85**, 036405 (2012) finds that TNSA proton acceleration in 2D and 3D simulations exhibit the same dynamics, but that proton energies in 3D are $\sim 1/2$ lower than in 2D. These 2D/3D comparisons are made at laser intensities of 1.38×10^{20} W/cm².

- d’Humières et al., Phys. Plasmas **20**, 023103 (2013) reaches a similar conclusion: 2D simulation give TNSA proton energies that are ~ 1.5 higher than in 3D. All these 2D/3D comparisons are made at laser intensities of 2×10^{20} W/cm².

A very important point relative to these two papers is that their simulations are conducted using very short pulse lasers (25-45 fs). In this case, **the protons reach the saturation of their energy only after the laser pulse has stopped**. This is clearly seen e.g. in Fig.C below. This means that the proton experience two phases during their acceleration: (1) an isothermal phase when the laser is on, followed by (2) an adiabatic phase when the lasers is stopped. **Is it precisely when the acceleration regime changes from isothermal to adiabatic that the difference between 2D and 3D appears**, as clearly appears in Fig.C below. This difference is due to that fact that there is a quicker adiabatic cooling of the sheath electrons in 3D than in 2D (and since the adiabatic cooling governs the dilution of the accelerating sheath field). In Fig.C, the deviation in proton energy between 2D and 3D clearly starts when the (45 fs duration) pulse has stopped.

Figure C: Fig. 6 of the paper of d’Humières et al. [Phys. Plasmas **20**, 023103 (2013)] showing that the deviation between 2D and 3D computed proton energies takes place, when using a 45 fs duration laser pulse in the simulation, only after the laser pulse has stopped. The deviation is obviously due to a difference in the adiabatic phase of the expansion which is here important because the protons have not yet reached their energy at saturation. This is not the case in our paper since the laser pulses we use are long (700 fs for the SNL experiment), hence the protons are fully accelerated during the laser pulse, and hence no deviation between 2D and 3D is expected.

In our case, the pulse duration used in the SNL experiment (performed at the highest intensity used in our study) is 800 fs. This is much longer than the pulse duration used in the above 2 papers. Our PICLS-2D simulation shows that the proton energy in our case saturates around 300 fs after the peak, which is still before the laser pulse ends. **This means that the protons in our experiment acquire their energy before the adiabatic cooling phase takes place**; the sheath field at the centre is maintained thanks to a continuous supply of fresh electrons. This is the reason why our 2D PIC simulations at high intensity fit well with our experimental

data, and that **we do not expect that 3D effects would induce a deviation from what is predicted in 2D.**

This said, doing real 3D PIC simulations in our conditions (long pulse duration and fairly large dimensions) is unfortunately still beyond our capability, but we believe that the argument detailed above and based on published papers is solid. As discussed in the manuscript, the deviation of the proton energy from the 1D model is due to the magnetic field. The 1D model intrinsically neglects the magnetic field and thus predicts unrealistically high proton energy at high-intensity.

This is now clarified as follows in the paper on p.10:

“For laser intensities below $10^{20} \text{ W} \cdot \mu\text{m}^2 \cdot \text{cm}^{-2}$, the consistency between the 1D analytical model, the 2D simulations and the experimental data indicates that the acceleration is quasi-1D, as already demonstrated in a number of studies [4,42,43]. For higher laser intensities, the progressive deviation between the 1D unmagnetized model on the one side, and the data and 2D simulations on the other side points to increasing multi-dimensional effects. This is consistent with the increasing deflections of the electrons and ions (highlighted in Fig.1 and Fig. 4) induced by B-fields of growing strength, which can be only modelled in a multidimensional geometry. Under our long-pulse conditions, we expect that 2D simulations capture the ion acceleration to relatively good accuracy since the protons should reach their saturation energy within the laser pulse, i.e. during the isothermal acceleration stage [19]. Indeed, as was shown in Refs [44,45], differences between 2D and 3D simulations mostly arise during the adiabatic acceleration stage (i.e. after the laser irradiation) for protons that have not already reached their final energy in the isothermal stage.”

3. The authors claim that the self-generated magnetic fields pose fundamental limit for TNSA. In fact, it is not a fundamental one, but instead is a consequence of a particular set of experimental conditions: tight focusing of a long laser pulse. As the authors discuss later in the paper, deleterious effects can be decreased by using short laser pulses and loose focusing.

We apologize for the lack of clarity and precision of our paper. What we mean is that the effect of the magnetic field highlighted in the present paper is a fundamental one, and one that was not, to our knowledge, revealed in previous studies. When the magnetic fields are strong, they will hamper proton acceleration, whatever are the other circumstances. And **strong magnetic field will always appear when increasing the laser intensity, at some point, even for extremely short laser pulses.**

We agree, and actually show, that the magnetic fields are not strong enough to be detrimental to proton acceleration in all conditions. This is for example the case for laser intensities below 10^{20} W/cm^2 . Ultra-short laser pulses also allow to circumvent the issue, provided that their intensity is not too strong. To verify that indeed using, at the same intensities that explored in the paper, shorter pulse lasers leads to lesser magnetic inhibition, we performed an additional PIC simulation (also reported now in Section 7 of the *Suppl. Info.*, see Fig S.9 there) using a short laser pulse (SLP), i.e. significantly shorter (50 fs) than the one used for the highest results shown in the main paper (700 fs).

In Figure D (same figure as Fig S.9 in the *Suppl. Info.*) are shown the results of that PIC simulations in the case of SLP (50 fs). The wavelength, the intensity and the spot size of the

laser pulse were kept equal to those used in Fig. 1, i. e. $1\ \mu\text{m}$, $2 \times 10^{21}\ \text{W}/\text{cm}^2$ and $1.6\ \mu\text{m}$ respectively. The duration at FWHM was equal to 50 fs. The target had the same structure and density as at the simulations shown in Fig. 1 of the main text. The simulations were performed using the PICADOR code, see Methods for details.

Figure D: 2D PIC simulations of proton acceleration by a 50 fs laser pulse (see text for details). Shown are (a) electron and (b) ion concentrations, (c) trajectories of sampled energetic electrons and (d) the transverse B -field (B_z) distribution. All snapshots are taken at the peak of the laser pulse.

When comparing the simulation results with 50 fs and 700 fs pulse durations, several points can be noticed:

- In the case of the 50 fs pulse, the generated magnetic field at the peak of the laser pulse is the same order.

- Similar electron energies are reached (which was rather expected since the laser intensities are the same).
- Similar maximum proton energies are attained, although a slightly higher value is found for a 50 fs pulse, consistent with experimental empirical scalings (see Fig. S.5 in the *Supplementary Information*).

Since in the 50 fs case, the laser intensity reaches its maximum much quicker, the protons reach their saturation energy over a much shorter distance, and hence the sheath longitudinal length is shorter than in the case of 700 fs duration. It is in fact even shorter than the electron Larmor radius over the timescale of proton acceleration, so that the electrons are weakly magnetized in the 50 fs case even though the magnetic field strength are similar to the case of 700 fs duration, as is obvious looking at Fig. D.c when compared with the Fig. 1 b) in the main text (see also the discussion in the section 6 in the *Suppl. Info.*). Yet it should be noticed that, as the magnetic field strength grows with the laser intensity, magnetic inhibition will eventually set in for intense enough pulses of a few 10 fs.

We also note that not all major existing and future facilities use ultra-short pulses. Take for example the planned large-scale L4 laser at ELI-Beamlines which has moderate duration of 100-150 fs [B. Rus et al., Proc. SPIE **8780**, 87801T (2013)].

Moreover, going to ultra-short pulses and loose focusing, as the reviewer rightly mentions, would come at a high cost: ultra-short pulses impose keeping the laser bandwidth all the way, with high costs for optics coating; loose focus means cranking the laser energy to keep a high intensity, which is also very costly. The aim of the community is to push toward table-top acceleration of ions. One of the best results so far (85 MeV) has been indeed obtained using a large-scale, moderately short laser (PHLIX). The design of future facilities geared toward optimizing ion acceleration will obviously have to integrate all these factors into account, and that is one of the messages of the paper.

We clarify this point in the paper with the following:

“Nonetheless, we expect that at very high intensities, even for the shortest pulse durations, the magnetic field strength will unavoidably become large enough that the magnetization effect highlighted here will start to be effective. This is why the B-field effects highlighted in the present paper might pose a fundamental limit to laser-based ion acceleration.”

4. Figure 2 presents results of simulations and one experimental datapoint, but the datapoint is taken from an experiment with loose focusing, while the simulations were performed with tight focusing. As it is shown by the authors, those two situations result in quite different outcomes, so I don't see how they can be put on one plot. Also, the datapoint should have an associated errorbar.

To clarify the influence of the laser spot size on the B-field strength, we have performed new calculations (which are detailed now in Section 6 of the *Suppl. Info.*). The results are summarized in Fig. E below.

If one compares between a tight focusing (1.6 μm) and a medium focusing (4 μm or 10 μm) at a given intensity, the loose focusing provides slightly higher B-field. This is due to the fact that, for a given hot electron divergence (a 45° half angle is used here) and target thickness

(2 μm), hot electron dilution occurs faster at smaller spot sizes. Hence, the initial electron density at the target rear gets lower in the tight focusing case, which leads to a weaker initial sheath electric field and a lower B-field. Only when the spot size becomes comparable to the sheath size (i.e. when it is much larger than 40 μm according to Figure E), does the B-field start to decrease. We note that such large spot is seldom used, except on very few large-scale facilities, e.g. the PETAL laser [A. Casner et al., HEDP **17**, 2–11 (2015)].

However, we should also note that all these differences in B-field strength for various laser spot sizes still remain in the same order of magnitude, hence the justification to plot them together. The difference between the 1.6 μm spot and 4 - 5 μm spot conditions is quantitatively small and doesn't change the message of the figure. Indeed, as our message is "the B-field increases with intensity, and potentially reach Giga-Gauss level at $>10^{21}$ W/cm²", this message is true for both spot size conditions.

In addition, the point raised by the reviewer actually triggered us to add to Fig.2 of the paper (and Fig.E.d below) a few more experimental measurements of B-field strength that we could find in the literature. Unfortunately, and due to the difficulty in performing such measurements, there are very few of these. They are now reported in Figure E (d) below with their respective error bars reported.

We added a small discussion on this point in the main manuscript, on p. 7-8:

"Note that these data are not all acquired using the same focal spot size, which is not the same used in the model, but as detailed in Section 6 of the *Suppl. Info.*, the dependence of the B-field strength on the laser focal spot at a given intensity is actually rather weak. This is due to the fact that the sheath transverse gradient (responsible for the B-field generation) is dominated by fast recirculation and transverse spread of the hot electrons before the ions have time to move. This holds as long as the laser spot size is smaller than the transverse sheath size (of the order of 40 μm , see Fig.1.a1 and b1 and Section 6 of the *Suppl. Info.*)."

Figure E Magnetic field strength dependence as a function of the laser spot size as predicted by the 1-D model. (a) Schematic drawing to show how the lateral spread of the hot electrons due to recirculation is estimated at $t = t_{char}$, yielding the sheath diameter, $2r$. (b) Characteristic time t_{char} when the plasma expansion starts, defined as the time when the acceleration front becomes larger than the local Debye length. (c) Sheath diameter at $t = t_{char}$ due to fast electron recirculation and transverse spread. (d) Magnetic field strength calculated for various laser spot sizes and laser peak intensities, taken at the laser intensity peak. For all curves, the laser wavelength is $\lambda_L = 1 \mu m$. The red circles correspond to experimental observations ([A] G. Sarri et al., *Phys. Rev. Lett.* **109**, 205002 (2012), [B] B. Albertazzi et al., *Phys. Plasmas* **22**, 123108 (2015), [C] W. Schumaker et al., *Phys. Rev. Lett.* **110**, 015003 (2013), [D] M. Tatarakis et al. *Nature* **415**, 280 (2002)), while the blue boxes come from the PIC simulations of the paper (see Fig.1 of the main text).

5. Given these questions regarding scientific soundness of both experimental and simulated results, I cannot recommend the paper for publication in Nature Communications.

With all the additional calculations and new simulations presented above, together with all the changes made to the article, we hope that we met the expectations of the reviewer, and that our article could be considered for publication.

Reviewer #3 (Remarks to the Author):

1. The authors claim that the B-field generation at the target rear surface hampers the TNSA ion energy.

To highlight their idea, the authors provide 2D PIC simulations and a couple of elementary formulas like calculating particle cyclotron radius and rough estimates for the B-field.

The only evidence they provide that the B-field influences the TNSA mechanism (of course it does - one has to solve corresponding MHD equations) - they compare 1D TNSA model with 2D PIC simulations.

First, let us clarify that **MHD cannot be a proper theoretical frame to describe the current phenomenon we are analyzing**. This is because its origin is essentially kinetic. First, the accelerating electric fields as well as the magnetic fields on the rear side of the target are produced by electron bunches ejected by the laser from the irradiated target side and breaking out on its rear side. The electrons in those bunches are energetic enough to not collide with other particles on the spatial scales of the sheath. Consequently, we have electron streams in collisionless plasmas, which, as is well known, cannot be treated hydrodynamically, i.e. in a MHD framework. Second, even in the case where the B-fields are strong enough to magnetize a large fraction of the electrons at the target interfaces, these fields vanish in the central (on-axis) region. Third, another key ingredient of MHD, namely quasi-charge neutrality, is violated at the ion front where the largest ion energies are reached. For the above reasons, we believe that MHD cannot be applied to the problem under consideration.

MHD, however, may be applied later when the laser pulse has ended up and the sheath starts to expand adiabatically. However, this part of the acceleration process doesn't have much influence on the phenomena discussed in our paper since at this stage the protons are already fully accelerated (as detailed in the answer to reviewer #2, see paragraph 2).

We would like also to stress that our paper is not an analytical model/simulation paper. The evidence for the B-field detrimental effect at very high intensities on proton acceleration is not based solely on the comparison between the 1D model and the 2D PIC simulations, but on experimental data compared to these models.

Although there have not been so many measurements of B-field in high-intensity interactions, there is now ample experimental evidence that strong B-fields exist, and that their strength increases with the laser intensity (see the augmented Fig.2 in the main manuscript, which is reproduced here as Figure F).

In our paper, we bring experimental evidence for:

- The progressive deviation (lowering) of the proton energy compared to the unmagnetized 1D model when we increase the laser intensity, and therefore the strength of the induced B-field.
- The magnetic deflection of the protons at very high laser intensity and the proton energy dependent ring diameter of accelerated proton patterns shown in Fig.4. As shown in Fig. S.4 in the supplementary information, the measured proton deflection angle suggests the generation of 300 – 400 Mega-Gauss (MG) B-field, consistent with our simple modeling and with the scaling of the B-field strength shown in Fig.F.a) (Fig.2.a) of the main text).

From these facts, and previous direct measurements of the B-fields, we believe there is no doubt on (1) the existence of magnetostatic fields of a few 100 MG at the target rear, and (2) their detrimental effect on proton acceleration. The latter point is demonstrated by computing the trajectories of the MeV electrons and protons under the action of the induced fields, by means of self-consistent PIC simulations (Fig. 1) as well as a simple particle tracing code (see Section 2 of the *Supplementary Information*).

Figure F Magnetic field strength and particle magnetization dependences against the laser pulse intensity, as predicted by the 1-D model. (a) Blue solid curve: magnetic field strength vs. laser intensity from the 1D model (details in the Methods. Laser parameter: $\phi_L=1.6 \mu\text{m}$, $\tau_L=700 \text{fs}$ and $\lambda_L=1 \mu\text{m}$) taken at 350 fs after the plasma expansion starts (i.e. corresponding to the temporal peak for the 700 fs pulse). The red points correspond to observations ([A] correspond to [Sarri et al. Phys. Rev. Lett. 109, 205002 (2012)], [B] correspond to [Albertazzi et al. Phys. Plasmas 22, 123108 (2015)], [C] correspond to [Schumaker et al. Phys. Rev. Lett. 110, 015003 (2013)], and [D] correspond to [Tatarakis et al. Nature 415, 280 (2002)]), while the blue boxes come from PIC simulations (see Fig.1 in the main text). (b) Normalized Larmor radius R_L/x_{front} at the laser intensity peak for both electrons (red) and protons (blue), as predicted by the 1D model used in Figs.1.a2-a3 and Figs.1.b2-b3. The laser parameters are those of Fig.1.b in the main text (i.e., $\phi_L=1.6 \mu\text{m}$, $\tau_L=700 \text{fs}$ and $\lambda_L=1 \mu\text{m}$).

2. The difference fails fairly into the statistical scatter. For example, "Exp. SNL direct" dot in Fig. 3 is factor 2 above the analytical trend, while "SNL EPM" data are factor 2 below the trend.

We now clarify the description of the figure (Fig.3 of the main text). Of course there is scatter of the data around the value given by our simple model, but the main point is that all the points below $I=10^{20} \text{ W/cm}^2$ are in the proximity of the 1D model, while the points at higher intensities are all well below the 1D simple trend that ignores the magnetic influence on the protons. And the gap increases with the laser intensity.

To make clear the issue of the SNL data, and its consistency with the 1D model at low intensity, as well as its deviation from the latter at high intensity, we added the analytical line for the "Exp. SNL direct" (in the previous version of the paper, we just showed the line corresponding to the tight/EPM focus condition of SNL) as well as "Exp. LULI direct". The calculation is done in exactly the same way as other analytical lines as described in the

Methods[†]. When increasing the laser focal spot size for a given laser intensity, due to the finite divergence of the electrons[‡], the target thickness and the increasing laser energy, the initial electron density at the target rear increases. This lead to higher sheath electric field, and thus to the prediction of higher proton energy by the model, as seen in the new Figure 3 in the main text (which is reproduced here as Figure G below) –compare the solid and dashed lines. In the figure, the "Exp. SNL direct" data point (red triangle) is shown to fit well with this analytical trend (the red dashed line). Also, the experimental data points at higher laser intensity appear to be well below the “SNL EPM” analytical trend-line, i.e., clearly outside the scatter (note that the four experimental data points exhibit little scatter).

Figure G *Maximum proton energy recorded in experiments as a function of the peak on-target peak intensity ($I_L \lambda_L^2$) and deviation from a 1D model neglecting magnetic field effects. For the LULI experiments (black or blue open symbols), the targets were Al and Au foils of thickness between $0.5 \mu\text{m}$ and $2 \mu\text{m}$ and of transverse dimensions $> 1 \times 1 \text{mm}^2$. For the SNL experiment (red open symbols), the targets were $1.1 \mu\text{m}$ thick Au foils with transverse dimensions between $50 \times 50 \mu\text{m}^2$ and $300 \times 300 \mu\text{m}^2$ (note that no trend on the size was observed). Each point corresponds to a single laser shot. Lines plot predictions of the 1D plasma expansion model considered in Fig.1 and Fig.2 in the main text (see Methods). Closed symbols represent PICLS-2D simulation results with $2 \mu\text{m}$ thick Al targets: closed circles and triangles are for $50 \times 50 \mu\text{m}^2$ wide targets and closed red squares for $20 \times 20 \mu\text{m}^2$ wide targets. The error bars on the proton energy quantify the energy range of the radiochromic films used to diagnose the TNSA protons. The error bars in the laser intensity arise from the uncertainty in estimating the encircled laser energy within the FWHM spot. (inset)*

[†] Note that, the model predicts slightly lower proton energy compared to what has been shown in our previous main manuscript Fig. 3. This is because we have extracted the proton energy at $t = 1.3\tau_L$ after starting the expansion (τ_L is the laser pulse duration in FWHM), while now we took at $t = \tau_L$. As according to the PICLS simulation, τ_L proton acceleration is already completed at $t = \tau_L$ for the case of laser pulse durations we explored here. This modification doesn't quantitatively change the message of the figure though.

[‡] Here we assumed 45 degree in half-angle divergence [Coury, M et al., Appl. Phys. Lett. 100, 074105 (2012)]

Experimental setup employing a refocusing ellipsoidal plasma mirror (EPM) to reduce the laser focal spot and increase the laser intensity.

This is now clarified in the text as follows, on p.10:

“Figure 3 summarizes the maximum proton energy recorded in these experiments (see open symbols) as a function of the peak laser intensity over three orders of magnitude, up to $1.3 \times 10^{21} \text{ W} \cdot \mu\text{m}^2 \cdot \text{cm}^{-2}$. The experimental data are fairly well reproduced by 2D PIC simulations (see closed symbols) performed in the same conditions. The solid lines plot the results of the 1D plasma expansion model employed in Fig.1 and Fig.2 which neglects magnetization effects. These analytical predictions reasonably agree with the experimental measurements, performed at LULI (blue symbols) and SNL (red symbols), at the lowest laser intensities. However, both 2D PIC simulations and experimental results gradually deviate from the model's predictions when the laser intensity exceeds $10^{20} \text{ W} \cdot \mu\text{m}^2 \cdot \text{cm}^{-2}$; compare the 1D model's solid red line vs the data (red open circles) and simulations (red closed circles) for intensities above $10^{21} \text{ W} \cdot \mu\text{m}^2 \cdot \text{cm}^{-2}$.”

3. In general, the influence of B-field on both electron dynamics and ion acceleration are doubtless.

The reviewer is right, in general. However, the effect of the B-field is noticeable on the particle dynamics **only at very high laser intensities**. This is what we mean to show in our paper since the B-field does not modify the particle dynamics when the laser intensity is below 10^{20} W/cm^2 .

Up to now, the laser-plasma community has overlooked B-field effects on the particle dynamics precisely because these effects were not significant at the laser intensities used in experiments. That is why we hardly find any hint or mention of this possible inhibition effect in the literature (expect a short mention in A. Pukhov, PRL **86**, 3562 (2001); “*The generated magnetic fields are of the order of 10 MG. This is enough to collimate the hot electrons and influence directionality of the plasma expansion. ... In our simulation with relatively thin plasma layers, we observe a ringlike angular distribution of the energetic ions...*”). But this simulation was done using much lower intensity and lower magnetic field compared to our investigations, and didn't discuss the detrimental effect played by the B-field on high-energy proton acceleration at very high intensity).

We also note that there have been innumerable papers published, most of them theoretical, on the way to optimize ion acceleration by lasers and on pathways to push the ion energy forward, and to the best of our knowledge none of them seems to mention that when pushing the laser intensity, such self-generated B-fields would grow to the point of effectively hampering ion acceleration. For us, this came in fact a bit as a surprise (the fact that the self-generated B-field would be strong enough to cause such an effect) when we analyzed our data as it was not a common idea in the community that B-fields would play such negative role. In fact, the realization that very high strength B-fields could be self-generated in these conditions emerged only quite recently, as the result of a handful of papers measuring these B-fields [M. Tatarakis et al. Nature **415**, 280 (2002), G. Sarri et al., PRL **109**, 205002 (2012), B. Albertazzi et al., *Phys. Plasmas* **22**, 123108 (2015), W. Schumaker et al., PRL **110**, 015003 (2013) – see the datapoints in Fig.2.a of the main text]

All this is now clarified as follows (the modified text is highlighted in yellow):

In the abstract:

“Here, we present experimental and numerical results demonstrating that magnetostatic fields self-generated on the target surface may pose a fundamental limit **to target normal sheath ion acceleration for high enough laser intensities.**”

Also on p.2, we precise:

“Most of these studies, however, have overlooked a potentially important factor: the feedback effect on the electrons and accelerating ions of magnetic (B) fields that are self-generated on the target surfaces **and can act detrimentally on the particle dynamics for high enough laser intensities.**”

4. Still, I am not convinced that the paper in this form satisfies all Nature conditions. May be, it should be published in a more specialized journal.

With the clarifications brought in answer to the comments of the reviewer, plus all the additional calculations and new simulations presented above, together with all the changes made to the article, we hope that we met the expectations of the reviewer, and that our article could be considered for publication.

Reviewers' comments:

Reviewer #1 (Remarks to the Author):

The authors have done a reasonable job responding to the critiques of the manuscript, which is substantially improved. The figures still seem like they are more suited for a presentation rather than a publication however (text does not show up well when I print it out). This should provide some thought provoking discussion into laser ion acceleration and tabletop accelerators as a whole. I am a bit unnerved about the authors no longer having their simulation data, however; one should never delete data particularly if they have not published the results of it. In any case, I think the manuscript is sufficiently improved to justify publishing.

Reviewer #2 (Remarks to the Author):

The authors successfully addressed all points I raised in my previous review. I believe the paper addresses an important issue and will be of interest to laser-driven ion acceleration community. I recommend the paper for publication.

Reviewer #3 (Remarks to the Author):

The authors discuss the influence of B-field on ion acceleration in the TNSA regime. I have read the rebuttal and still do not see why the MS has to be highlighted in Nature. The ultra-strong B-field generation in laser-plasma experiments is well known. Its strength is comparable with that of the TNSA electric field. Certainly, the B-field has to be taken into account when considering ion acceleration. This is evident and more or less trivial.

The good question is how it should be taken into account. Presently, the authors show some experimental blobs and 2D PIC simulations in slab geometry. They add some doubtful "analytics" that looks more like philosophic conjecture with no true check. This is by far insufficient to identify the correct mechanism.

The authors claim, the MHD is inapplicable here because "all the effects are kinetic". Yet, at the 100 MG strength of the B-field, the electrons are strongly magnetized: 1 MeV electron Larmor radius is 0.2 μm only. This makes the MHD description very reliable. It is the MHD dynamics that drives the foil after the laser pulse is over.

Finally, I suggest, this paper be published in a more specialized journal like Physics of Plasma, or comparable, where experts can discuss the results.

Answers to the Reviewers' comments:

Reviewer #1 (Remarks to the Author):

1. The authors have done a reasonable job responding to the critiques of the manuscript, which is substantially improved. The figures still seem like they are more suited for a presentation rather than a publication however (text does not show up well when I print it out). This should provide some thought provoking discussion into laser ion acceleration and tabletop accelerators as a whole. I am a bit unnerved about the authors no longer having their simulation data, however; one should never delete data particularly if they have not published the results of it. In any case, I think the manuscript is sufficiently improved to justify publishing.

We thank the reviewer for his/her encouraging comments. We paid a lot of attention to making the figures clear and readable, but there are obviously still issues. Therefore, **we have now enlarged the font size in all the figures** in order to improve their readability and printability.

Regarding the issue of the simulations storage, we are not sure if there is an established common practice, but at least for the present authors of the paper who have been performing the numerical simulations (Y. Sentoku from Osaka U., A. Khorzimanov from IAP-RAS, and L. Gremillet from CEA-DIF), we very rarely keep full simulation results as each run can represent Tb of data, and at present market costs, would induce astronomical costs in storage devices. Our common practice is rather to analyse the numerical runs, keep only a small part of the data, and then free the storage space for new simulations. It is also to be mentioned that the person (Y.S.) in charge of the PICLS simulations recently moved to Osaka U. from UNR, further complicating the preservation of the whole simulation data.

However, as shown by the excellent comparison, using the same initial parameters, of the predictions of the PICADOR or the PICLS codes, we do not see issues in not having the full data of a particular run, which can be re-run and will yield the same results. It is in fact because the results have been shown to be robust to the choice of a particular code and to the precise choice of initial conditions that we believe that the physics highlighted here is right.

Reviewer #2 (Remarks to the Author):

1. The authors successfully addressed all points I raised in my previous review. I believe the paper addresses an important issue and will be of interest to laser-driven ion acceleration community. I recommend the paper for publication.

We thank the reviewer for his/her encouraging comments.

Reviewer #3 (Remarks to the Author):

1. The authors discuss the influence of B-field on ion acceleration in the TNSA regime. I have read the rebuttal and still do not see why the MS has to be highlighted in Nature. The ultra-strong B-field generation in laser-plasma experiments is well known. Its strength is comparable with that of the TNSA electric field. Certainly, the B-field has to be taken into account when considering ion acceleration. This is evident and more or less trivial.

Although the generation of ultra-strong B-fields in intense laser-plasma interactions has been observed numerically about twenty years ago, **it is still not well understood, and its implications for ion acceleration have been essentially overlooked.** Compared to other aspects of intense laser-plasma physics, this topic has been relatively little addressed experimentally [besides the four papers cited in Fig.2.a of the main text, one should mention those from Indian labs using table-top lasers: A. Sandhu *et al.*, Phys. Rev. Lett. **89**, 225002 (2003); Phys. Rev. E **73**, 036409 (2006); S. Kahaly *et al.*, Phys. Plasmas

16, 043114 (2009); S. Mondal *et al.*, PNAS **109**, 8011 (2012)], and this is due, we believe, to the difficulty of measuring these B-fields. But if the issue is so obvious, one could wonder why the evidence for these fields has been highlighted in a number of high-impact journals. This is not evidencing, at least according to the editors of these journals having undoubtedly wide audiences, for a topic having little impact in the broad physics (and beyond) community.

Let us stress again that the present work is the first that addresses, both experimentally and numerically, the potentially deleterious effects of surface B-fields on TNSA proton acceleration at ultra-high laser intensity. Such fields were briefly mentioned in the paper by A. Pukhov [Phys. Rev. Lett. **86**, 3562 (2001)], yet only to account for a ‘ring-like’ pattern in the proton profile observed in PIC simulations run at much lower laser intensities (10^{19} W. cm⁻²) and electron densities ($10n_e$) than under our conditions. In addition to providing experimental and numerical evidence for the magnetic inhibition of TNSA, we have developed a simple model that captures the salient features of the problem, and delineates the threshold parameters for this inhibition. We strongly believe that our experimental and numerical findings are of major significance for the understanding of laser-based ion acceleration on the forthcoming multi-PetaWatt laser facilities.

2. The good question is how it should be taken into account. Presently, the authors show some experimental blobs and 2D PIC simulations in slab geometry. They add some doubtful "analytics" that looks more like philosophic conjecture with no true check. This is by far insufficient to identify the correct mechanism.

We agree that our analytical calculations are very simplified, particularly in that they neglect the feedback of the self-generated fields on the particle dynamics. Our main goal, however, is to raise the issue of magnetic inhibition based on our state-of-the-art experimental results performed under rarely-attained laser conditions, thanks to our recent innovation in plasma focusing optics. Furthermore, most of our model predictions are compared with 2D full PIC simulations, so that **we strongly object to the Referee’s assertion that our work lacks ‘true check’**.

The authors claim, the MHD is inapplicable here because "all the effects are kinetic". Yet, at the 100 MG strength of the B-field, the electrons are strongly magnetized: 1 MeV electron Larmor radius is 0.2 μ m only. This makes the MHD description very reliable. It is the MHD dynamics that drives the foil after the laser pulse is over.

Here, we absolutely do not understand the reviewer’s point of view. That we know of, there have been no reported modelling of ion acceleration induced by intense (i.e. relativistic at the electron energy scale) lasers using MHD models. This is due to a **principle problem**. The overall and, we believe, widely accepted physics frame of ion acceleration in these conditions is the one of the acceleration being induced by a violent charge separation between the electrons and the ions following the energization of the electrons at $>MeV$ energies, while the ions stay initially at rest due to their inertia. This frame has been initially proposed by A. Gurevich in a quite famous paper [A.V. Gurevich *et al.*, Sov. Phys. JETP **22**, 449 (1966)], and then revisited in the modern context of high-intensity lasers, most famously by P. Mora [Phys. Rev. Lett. **90**, 185002 (2003)]. This model is, to our knowledge, not debated, and is presented as such in all reviews of ion acceleration by lasers published in well-recognized journals [see e.g. A. Macchi *et al.*, Rev. Mod. Phys. **85**, 751 (2013), or H. Daido *et al.* [Rep. Prog. Phys. **75**, 056401 (2012)].

Major issues in modelling laser-driven TNSA are **charge-separation effects at the ion front, electron kinetics** (i.e., the departure from a Boltzmann distribution function, and the self-consistent electron cooling dynamics) and, of course, multidimensional effects (and hence the interplay of electric and magnetic fields). Now, MHD is based on the assumptions of quasi-neutrality, small particle Larmor radii and thermal particle distributions. While part of these assumptions may locally hold *off axis* and *inside* the

expanding electron-ion plasma (where quasi-neutrality holds, the B-field is at its strongest, and the electrons may gyrate with radii smaller than the plasma/field scale-lengths), **MHD is clearly invalid at the ion front** (where the quasi-neutrality assumption breaks down) and/or around the axis (where the Bfield weakens and/or changes sign), that is, in the regions where the maximum ion energies are to be found. This is partially illustrated by the sample electron trajectories of Figs.1.a4, 1.b4 and S.10.c.: while some off-axis electrons exhibit drift motions typical of a magnetized plasma, those crossing the axis are clearly unmagnetized. In summary, we absolutely do not understand the reviewer's statement about the applicability of MHD to the present problem.

3. Finally, I suggest, this paper be published in a more specialized journal like Physics of Plasma, or comparable, where experts can discuss the results.

We thank the first two reviewers for their support to our findings. We cannot comment further to this point made by reviewer #3.

REVIEWERS' COMMENTS:

Reviewer #1 (Remarks to the Author):

The authors have responded to all comments and the paper is suitable for publication. I appreciate the efforts they took in increasing readability and responding to the comments with such care.

Reviewer #3 (Remarks to the Author):

Well, it seems that the discussion with the authors becomes philosophical: whether the MHD can describe the target thermal expansion as observed by the authors in 3D.

The answer is: yes, may be excluding the thin Debye layer at the very leading front, where the interaction is kinetic indeed.

What the authors awkwardly describe as a "magnetic inhibition" or "magnetic deflection" is nothing else but the old magnetic pressure. Correspondingly, the authors have to compare the thermal pressure of their hot plasma and the magnetic pressure. In the low density expanding plasma at the rear of the target, where the temperature drops fast due to the adiabatic expansion, the B-pressure may indeed become dominant, what they see as the "magnetic inhibition".

It is the weird terminology of the authors and the awkward description of a probably well defined phenomenon that makes me unhappy about seeing this MS published in Nature. A more specialized journal would fit this paper much better.

Further, the simulations are 2D only, that is not the state-of-the-art today and I doubt it fits the high Nature standards.

Answers to the Reviewers' comments:

Reviewer #1 (Remarks to the Author):

The authors have responded to all comments and the paper is suitable for publication. I appreciate the efforts they took in increasing readability and responding to the comments with such care.

We thank the reviewer for his/her positive view of our work.

Reviewer #3 (Remarks to the Author):

1. Well, it seems that the discussion with the authors becomes philosophical: whether the MHD can describe the target thermal expansion as observed by the authors in 3D.

The answer is: yes, may be excluding the thin Debye layer at the very leading front, where the interaction is kinetic indeed.

In order to clarify for all readers the principle difference between MHD and PIC, and their respective ability to model the issue raised in the paper (the influence of self-generated magnetic field on the acceleration of non-thermal ion population driven by hot electrons), we have added the following text to the Methods section (on p.19, the modified text is highlighted in yellow):

“Numerical simulations

To analyze the plasma dynamics at play during and following the intense laser irradiation, we resort to particle-in-cell (PIC) numerical codes, which provide a first-principles simulation framework more adequate than magneto-hydrodynamics (MHD) codes. Indeed, MHD is based on the assumptions of quasi-neutrality, small particle Larmor radii and thermal particle distributions. While part of these assumptions may locally hold off axis and inside the expanding electron-ion plasma (where quasi-neutrality holds, the B-field is at its strongest, and the electrons may gyrate with radii smaller than the plasma/field scale-lengths), MHD is invalid at the ion front (where the quasi-neutrality assumption breaks down) and/or around the axis (where the B-field weakens and/or changes sign), that is, in the regions where the maximum ion energies are to be found.

The 2D PIC numerical simulations presented in this work are performed using the PICLS and PICADOR codes.”

2. What the authors awkwardly describe as a "magnetic inhibition" or "magnetic deflection" is nothing else but the old magnetic pressure. Correspondingly, the authors have to compare the thermal pressure of their hot plasma and the magnetic pressure. In the low density expanding plasma at the rear of the target, where the temperature drops fast due to the adiabatic expansion, the B-pressure may indeed become dominant, what they see as the "magnetic inhibition".

It is the weird terminology of the authors and the awkward description of a probably well defined phenomenon that makes me unhappy about seeing this MS published in Nature. A more specialized journal would fit this paper much better.

In order also to clarify the difference between the kinetic effects investigated in the paper, and that determine the properties of the most energetic ions that are accelerated by the hot electrons, and the thermal pressure effect on slower ions, we have added to the text (on p.7) the following precision:

“Another expected detrimental effect of the B-field is the inward force exerted by the **fluid-like** magnetic pressure on the target surface, which tends to counteract the accelerating TNSA field.”

3. Further, the simulations are 2D only, that is not the state-of-the-art today and I doubt it fits the high Nature standards.

To our best knowledge, there are only two papers as of yet reporting 3D PIC simulations (in very limited setups) of ion acceleration. These are references 44 and 45 of our paper. Thus, to state that performing 2D simulations does not correspond to the state-of-the-art (i.e. meaning that 3D is the state-of-the-art) does not seem really true to us.